# Transient structural variations have strong effects on quantitative traits and reproductive isolation in fission yeast

Daniel C. Jeffares[1,2,†], Clemency Jolly[1], Mimoza Hoti[1], Doug Speed[2], Liam Shaw[1,2], Charalampos Rallis[1,2,†], Francois Balloux[1,2], Christophe Dessimoz[1,3,4,5], Jürg Bähler[1,2] & Fritz J. Sedlazeck[6]

Large structural variations (SVs) within genomes are more challenging to identify than smaller genetic variants but may substantially contribute to phenotypic diversity and evolution. We analyse the effects of SVs on gene expression, quantitative traits and intrinsic reproductive isolation in the yeast *Schizosaccharomyces pombe*. We establish a high-quality curated catalogue of SVs in the genomes of a worldwide library of *S. pombe* strains, including duplications, deletions, inversions and translocations. We show that copy number variants (CNVs) show a variety of genetic signals consistent with rapid turnover. These transient CNVs produce stoichiometric effects on gene expression both within and outside the duplicated regions. CNVs make substantial contributions to quantitative traits, most notably intracellular amino acid concentrations, growth under stress and sugar utilization in wine-making, whereas rearrangements are strongly associated with reproductive isolation. Collectively, these findings have broad implications for evolution and for our understanding of quantitative traits including complex human diseases.

[1] Department of Genetics, Evolution and Environment, University College London, London WC1E 6BT, UK. [2] UCL Genetics Institute, University College London, London WC1E 6BT, UK. [3] Department of Computer Science, University College London, London WC1E 6BT, UK. [4] Department of Ecology and Evolution and Center for Integrative Genomics, University of Lausanne, Biophore, Lausanne 1015, Switzerland. [5] Swiss Institute of Bioinformatics, Biophore, Lausanne 1015, Switzerland. [6] Department of Computer Science, Johns Hopkins University, Baltimore, Maryland 21218, USA. † Present addresses: Department of Biology, University of York, York YO105DD, UK (D.C.J.); School of Health, Sport and Biosciences, University of East London, London E15 4LZ, UK (C.R.). Correspondence and requests for materials should be addressed to C.D. (email: christophe.dessimoz@unil.ch) or to J.B. (email: j.bahler@ucl.ac.uk) or to F.J.S. (email: fritz.sedlazeck@jhu.edu).

A variety of genetic changes can influence the biology of species, including single-nucleotide polymorphisms (SNPs), small insertion-deletion events (indels), transposon insertions and large structural variations (SV). SVs, including deletions, duplications, insertions, inversions and translocations, are the most difficult to type and consequently the least well described.

Nevertheless, it is clear that SVs have strong effects on various biological processes. Copy number variants (CNVs) in particular influence quantitative traits in microbes, plants and animals, including agriculturally important traits and a variety of human diseases[1–5]. Inversions are known to influence reproductive isolation[6–13] and other evolutionary processes such as recombination[8] and hybridization between species[14], with a variety of consequences[15].

We and others have recently begun to develop the fission yeast Schizosaccharomyces pombe as a model for population genomics and quantitative trait analysis[6,7,16–18]. This model organism combines the advantages of a small, well-annotated haploid genome[19], abundant tools for genetic manipulation and high-throughput phenotyping[20], and considerable resources of genome-scale and gene-centric data[21–23].

Previous analyses of fission yeast have begun to describe both naturally occurring and engineered inversions and reciprocal translocations[6,7,18]. Given this evidence for SVs and their effects in this model species, we recognized that a systematic survey of SVs would advance our understanding of their biological influence. Here, we utilize the recent availability of 161 fission yeast genomes and extensive data on quantitative traits and reproductive isolation[17] to describe the nature and effects of SVs in S. pombe.

We show that SVs have strong effects on a variety of quantitative traits and intrinsic reproductive isolation. They contribute an average of 11% of trait variance (the much more abundant SNPs contribute 24% on average), with the largest effects coming from CNVs. We show that CNVs are transient within clonal populations, and are frequently not well tagged by SNPs. We also show that rearrangements (inversions and translocations) contribute to reproductive isolation, whereas CNVs do not.

## Results

**Population-wide detection of structural variations.** To predict an initial set of SVs, we applied four inference software packages (Delly, Lumpy, Pindel and cn.MOPs)[24–27] to existing short-read data[17], using parameters optimized on simulated data (Methods). We then filtered these initial predictions, accepting SVs detected by at least two callers, to obtain 315 variant calls (141 deletions, 112 duplications, 26 inversions and 36 translocations). We release this pipeline as an open-source tool called SURVIVOR (Methods). To ensure a high specificity, we further filtered the 315 variants by removing SV calls whose breakpoints overlapped with low complexity regions or any that corresponded to previously annotated long terminal repeats (LTRs)[17]. Finally, we manually vetted all the remaining SVs by visual inspection of read alignments in multiple strains for all remaining candidates. This meticulous approach aimed to ensure a high-quality call set, to mitigate against the high uncertainty associated with SV calling[25].

This curation produced a set of 113 SVs, comprising 23 deletions, 64 duplications, 11 inversions and 15 translocations (Fig. 1a). Reassuringly, when applying our variant calling methods to an engineered knockout strain, we correctly identified the known deletions and called no false positives. Attempts to validate all rearrangements by PCR and BLAST searches of

de novo assemblies positively verified 76% of the rearrangements, leaving only a few PCR-intractable variants unverified (see Methods for details).

Most SVs were present at low frequencies, with 28% discovered in only one of the strains analysed (Fig. 1b). The deletions were generally slightly smaller (median length 14 kb, Fig. 1c) than duplications (median length of 21 kb), with the largest duplication extending to 510 kb and covering 200 genes (a singleton in strain JB1207/NBRC10570). The majority of CNVs were present in copy numbers varying between 0 and 16 (subsequently we refer to amplifications of two or more copies as 'duplications').

All SVs, particularly deletions and duplications, were biased toward the ends of chromosomes (Fig. 1d and Supplementary Figs 1 and 2), which are characterized by high genetic diversity, frequent transposon insertions and a paucity of essential genes[17], similar to Saccharomyces cerevisiae and Sa. paradoxus[28,29]. All SVs preferentially occurred in positions of low gene density and were strongly under-enriched in essential genes (Supplementary Fig. 2).

To describe SVs further, we conducted gene enrichment analysis with the AnGeLi tool (Supplementary Table 1), which interrogates gene lists for functional enrichments using multiple qualitative and quantitative information sources[30]. The CNV-overlapping genes were enriched for caffeine/rapamycin induced genes and genes induced during meiosis ($P = 4 \times 10^{-7}$ and $1 \times 10^{-5}$, respectively); they also showed lower relative RNA polymerase II occupancy and were less likely to contain genes known to produce abnormal cell phenotypes ($P = 1.8 \times 10^{-5}$ and $3 \times 10^{-5}$, respectively). These analyses are all broadly consistent with a paucity of CNVs in genes that encode essential mitotic functions. Rearrangements disrupted only a few genes and showed no significant enrichments.

**Duplications are transient within clonal populations.** Our previous work identified 25 clusters of near-clonal strains, which differed by <150 SNPs within each cluster[17]. We expect that these clusters reflect either repeat depositions of strains differing only at few sites (for example, mating-type variants of reference strains $h^{90}$ and $h^-$ differ by 14 SNPs) or natural populations of strains collected from the same location. Such 'clonal populations' reflect products of mitotic propagation from a very recent common ancestor, without any outbreeding. We therefore expected that SVs should be largely shared within these clonal populations.

Surprisingly, our genotype predictions indicated that most SVs present in clonal populations were segregating, that is, were not fixed within the clonal population (68/95 SVs, 72%). Furthermore, we observed instances of the same SVs that were present in two or more different clonal populations that were not fixed within any clonal population. These SVs could be either incorrect allele calls in some strains, or alternatively, recent events that have emerged during mitotic propagation. To distinguish between these two scenarios, we re-examined the read coverage of all 49 CNVs present within at least one clonal population. Since translocations and inversions were more challenging to accurately genotype, we did not re-examine these variants. This analysis verified that 40 out of these 49 CNVs (37 duplications, three deletions) were clearly segregating within at least one clonal cluster (Supplementary Fig. 3). For example, one clonal population of seven closely related strains, collected together in 1966 from grape must in Sicily, have an average pairwise difference of only 19 SNPs (diversity $\pi = 1.5 \times 10^{-6}$). Notably, this collection showed four non-overlapping segregating duplications (Fig. 2c, yellow highlight). This striking finding

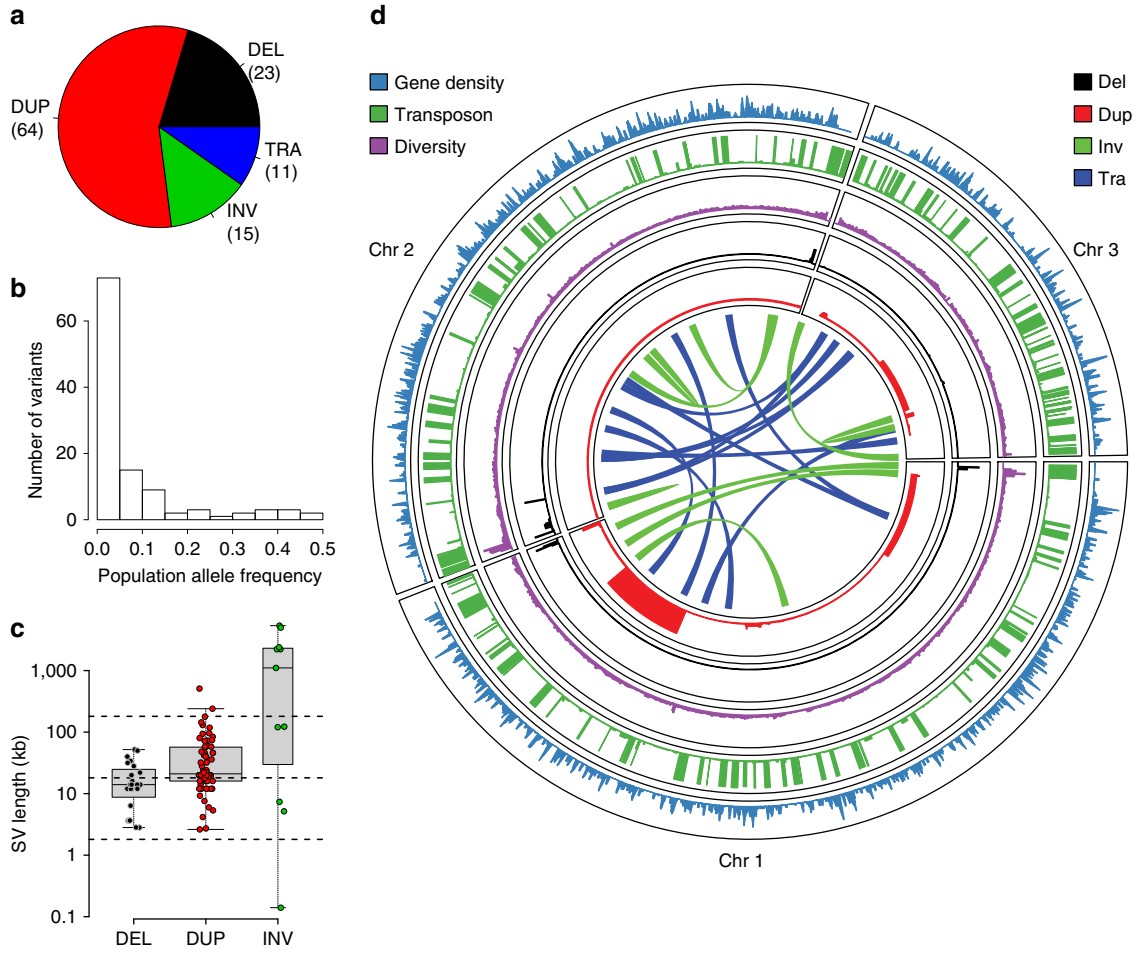

**Figure 1 | Characteristics of SVs in *S. pombe*. (a)** Relative proportions of SVs identified. Duplications (DUP) were the most abundant SVs, followed by deletions (DEL), inversions (INV) and translocations (TRA). **(b)** Population allele frequency distribution of SVs, showing the frequencies of less abundant alleles in the population (minor allele frequencies). **(c)** Length distributions of SVs, $\log_{10}$ scale. Deletions were smallest (2.8–52 kb), duplications larger (2.6–510 kb) and inversions often even larger, spanning large portions of chromosomes (0.1 kb–5,374 kb, see **d**). Horizontal dotted lines show the size of chromosome regions that contain an average of 1, 10 and 100 genes in this yeast. Box plots indicate the first quartile, the median and the third quartile; whiskers extend to the most extreme data point, which is no more than 1.5× the interquartile range from the box. **(d)** Locations of SVs on the three chromosomes compared with other genomic features. From outside: density of essential genes, locations of *Tf*-type retrotransposons, diversity ($\pi$, average pairwise diversity from SNPs), deletions (black), duplications (red) and breakpoints of inversions and translocations as curved lines inside the concentric circles (green and blue, respectively). Bar heights for retrotransposons, deletions and duplications are proportional to minor allele frequencies. Diversity and retrotransposon frequencies were calculated from 57 non-clonal strains as described by Jeffares *et al.*[17]

suggests that CNVs can arise or disappear frequently during evolution.

To examine whether this transience is a general feature of CNVs in this population, we quantified the variation in copy number of each CNV relative to mutations in the adjacent region of the genome. If a CNV was subject only to the same processes as these adjacent regions, we would expect a strong correlation between the rate of point mutation (SNPs) in these regions and the total variation in copy number of the CNV. However, the variation in copy number of CNVs across the data set was only weakly correlated with SNP variation in nearby regions of the genome (Spearman rank correlation $\rho = 0.22$, $P = 0.041$), indicating that CNVs are subject to additional or different evolutionary processes (Fig. 2a). Furthermore, some CNVs showed high rates of variation within closely related clusters relative to their variation in the rest of the data set (Fig. 2b,c, Supplementary Table 2 and Supplementary Fig. 4). Finally, we found that many CNVs represented the rare allele within the cluster, consistent with events that have short half-lives (Supplementary Fig. 5). Taken together, these results

indicate that CNVs are transient and variable features of the genome, even within extremely closely related strains.

**Transient duplications affect gene expression.** Partial aneuploidies of 500–700 kb in the *S. pombe* reference strain are known to alter gene expression levels within and, to some extent, outside of the duplicated region[31]. The naturally occurring duplications described here are typically smaller (median length: 21 kb), including an average of 6.5 genes. To examine whether naturally occurring CNVs have similar effects on gene expression, we examined eight pairs of closely related strains (<150 SNPs among each pair) that contained at least one unshared duplication (Fig. 3 and Supplementary Table 3). Several of these strain pairs have been isolated from the same substrate at the same time, and all pairs are estimated to have diverged ~50–65 years ago (Supplementary Table 3). We assayed transcript expression from log phase cultures using DNA microarrays, each time comparing a duplicated to a non-duplicated strain from within the same

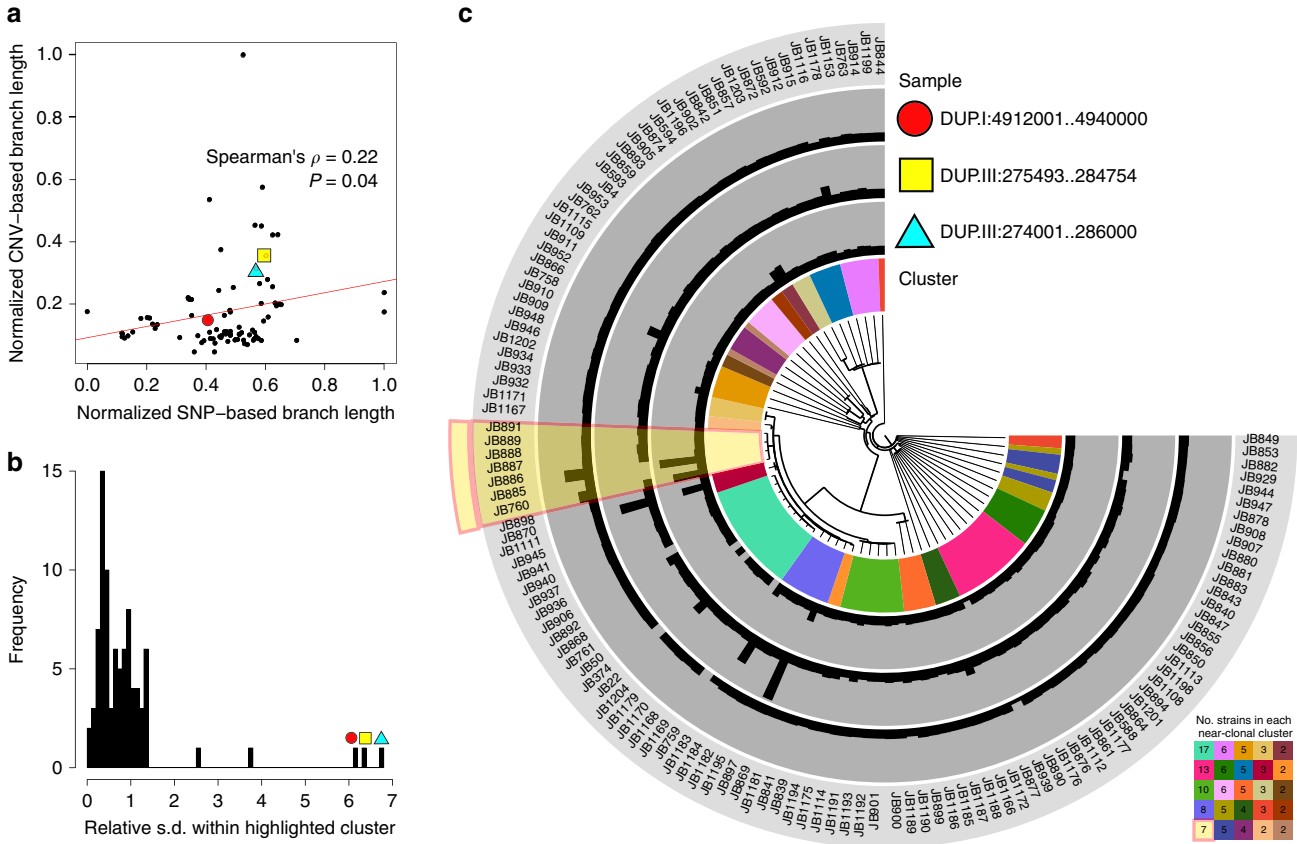

**Figure 2 | CNVs are transient within fission yeast.** (**a**) For each of the 87 CNVs we calculated the genetic distance between strains using SNPs in the region around the CNV (20 kb up- and downstream of the CNV, merged) as the total branch length from an approximate maximum-likelihood tree (*x* axis, SNP-based branch length normalized to maximum value). We further calculated a CNV-based distance using the total branch length from a neighbour-joining tree constructed from Euclidean distances between strains based on their copy numbers (*y* axis, CNV-based branch length normalized to maximum value). The weak correlation indicates that CNVs are subject to additional or different evolutionary processes. (**b**) Histogram of the standard deviation of each CNV within a near-clonal cluster (see also Fig. 2a), relative to its standard deviation across strains not in the near-clonal cluster. Standard deviation is highly correlated with CNV-based branch length (Spearman rank correlation $\rho = 0.90$, $P < 0.001$) (Supplementary Fig. 4b). The highlighted CNVs have unusually high rates of variation within this cluster compared with other clusters. (**c**) Copy number variation of these highlighted CNVs plotted on a SNP-based phylogeny (20 kb up- and downstream of the DUP.III:274001..286000 CNV) shows their relative transience within the cluster, as well as their variation across other near-clonal clusters. SNP-based phylogenies for the other two selected CNVs also do not separate the strains with different copy numbers (individual plots for each CNV across clusters for its corresponding SNP-based phylogeny are available as Supplementary data).

clonal population. In seven out of the eight strain pairs, the expression levels of genes within duplications were significantly induced, although the degree of expression changes between genes was variable (Fig. 3c and Supplementary Fig. 6). The increased transcript levels correlated with the increased genomic copy numbers, so that higher copy numbers produced correspondingly more transcripts (Spearman rank correlation $\rho = 0.71$, $P = 0.014$, Supplementary Fig. 7). No changes in gene expression were evident immediately adjacent to the duplications (Supplementary Fig. 7), suggesting that the local chromatin state was not strongly altered by the CNVs. This result not only confirms the previous observation that CNVs alter the gene expression levels, but more importantly it reveals large copy number differences between two genomes that are only 19 SNPs apart.

Interestingly, some genes outside the duplicated regions also showed altered expression levels (Fig. 3d and Supplementary Table 4). For example, two strain pairs differ by a single 12 kb duplication. Here, five out of seven genes within the duplication showed induced expression, while 45 genes outside the duplicated region also showed consistently altered expression levels (38 protein-coding genes, seven noncoding

RNAs) (Fig. 3d, arrays 7 and 8). As environmental growth conditions were tightly controlled, these changes in gene expression could be due to either compensatory effects of the initial perturbation caused by the 12 kb duplication or changes that arise due to SNPs or indels that segregate between the strains (Supplementary Fig. 6). We conclude that these evolutionary unstable duplications reproducibly affect the expression of distinct sets of genes and thus have the potential to influence cellular function and phenotypes.

**Copy number variants contribute to quantitative traits.** To test whether SVs affect phenotypes, we examined the contributions of SNPs, CNVs and rearrangements to 228 quantitative traits (Supplementary Table 5), including 20 cell-shape parameters, colony size on solid media assaying 42 stress and nutrient conditions[17], 126 growth parameters in liquid media conditions[7] and three biochemical parameters from wine fermentation[32]. For each phenotype, we used mixed model analysis to estimate the total proportion of variance explained by the additive contribution of genomic variants (the narrow-sense heritability).

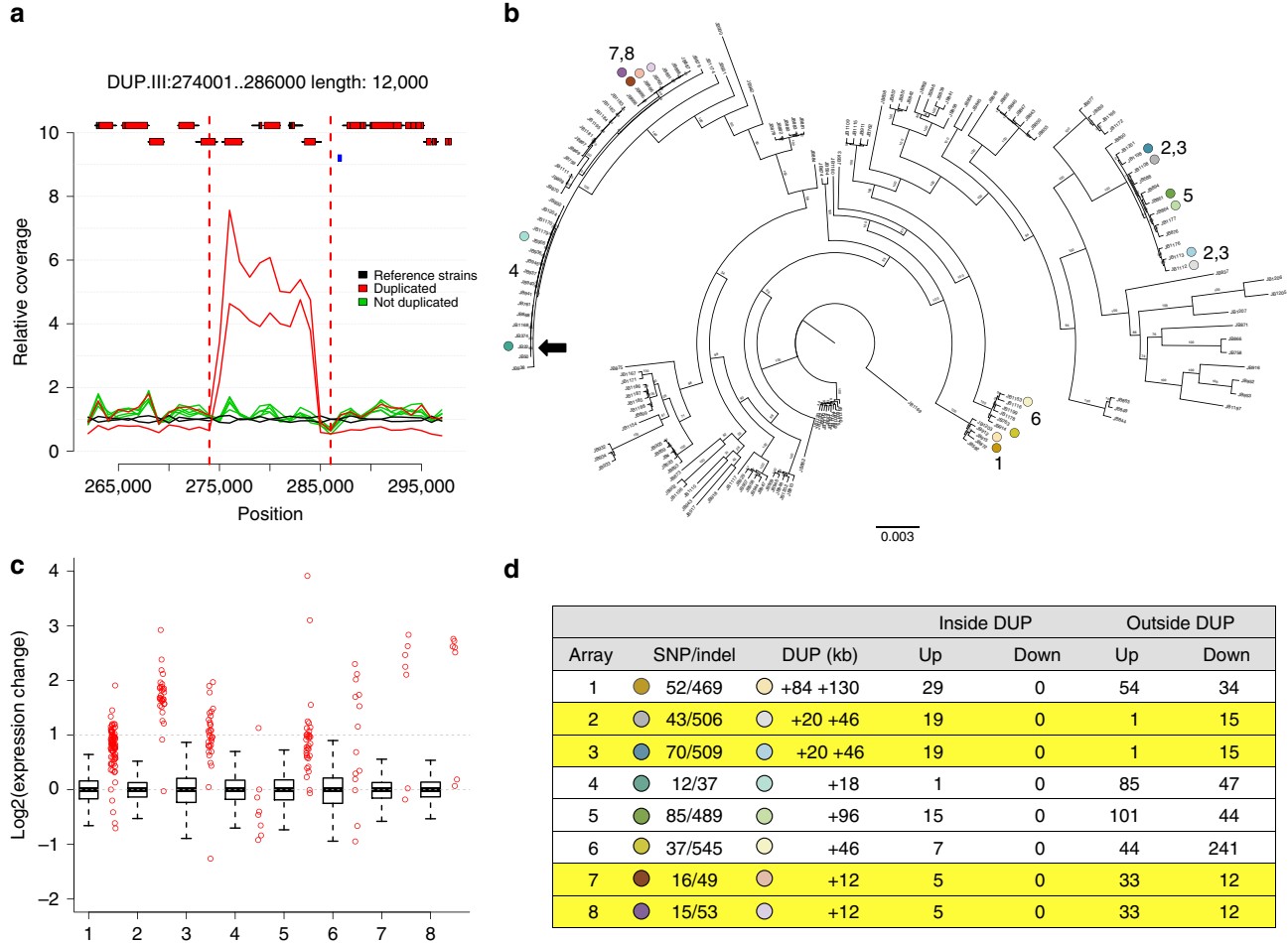

**Figure 3 | Transient duplications affect gene expression.** (**a**) Duplications occur within near-clonal strains. Plot showing average read coverage in 1 kb windows for two clonal strains (JB760, JB886) with the duplication (red), five strains without duplication (green) and two reference strains (h$^+$, and h$-$) (black). Genes (with exons as red rectangles) and retrotransposon LTRs (blue rectangles) are shown on top (see Supplementary Table 3 for details). (**b**) Eight pairs of closely related strains, differing by one or more large duplications, selected for expression analysis. The tree indicates the relatedness of these strain pairs (dots coloured as in **d**). The position of the reference strain (Leupold's 972, JB22) is indicated with a black arrow. The scale bar shows the length of 0.003 insertions per site. (**c**) Gene expression increases for most genes within duplicated regions. For each tested strain pair, we show the relative gene expression (strains with duplication/strains without duplication) for all genes outside the duplication (as boxplot) and for all genes within the duplication (red strip chart). In all but one case (array 4), the genes within the duplication tend to be more highly expressed than the genes outside of the duplication (all Wilcoxon rank sum test P values $<1.5 \times 10^{-3}$). Box plots indicate the first quartile, the median and the third quartile; whiskers extend to the most extreme data point, which is no more than 1.5 × the interquartile range from the box. (**d**) Summary of expression arrays 1–8, with strains indicated as coloured dots (as in **b**), showing number of SNP differences between strains, sizes of duplications in kb (DUP, where '$+ X + Y$' indicates two duplications with lengths $X$ and $Y$, respectively). We show total numbers of induced (up) and repressed (down) genes, both inside and outside the duplicated regions. Arrays 2,3 and 7,8 (in yellow shading) are replicates within the same clonal population that contain the same duplications, so we list the number of up- and downregulated genes that are consistent between both arrays. See Supplementary Tables 3 and 4 for details.

When we determined heritability using only SNP data, estimates varied between 0 and 74% (median 30%). After adding CNVs and rearrangements to SNPs in a composite model, the estimated overall heritability increased for nearly all traits, explaining up to ~40% of additional trait variance (Fig. 4a). This finding indicates that the CNVs and rearrangements can explain a substantial proportion of the trait variance. Using this composite model, we quantified the individual contributions of heritability best explained by SNPs, CNVs and rearrangements (Fig. 4b). On average, SNPs explained 24% of trait variance, CNVs 7% and rearrangements 4% (Supplementary Table 5). Analysis of simulated data confirmed that the contribution of CNVs could not be explained by linkage to causal SNPs alone (Supplementary Fig. 8).

Many trait measures gathered using the same method (for example, growth on solid media, cell shape) are strongly correlated[17]. Thus, some groups of traits have consistently larger contributions from SVs (Fig. 4b) than from SNPs alone. These traits include intracellular amino acid concentrations, growth under stress and several traits measured during wine fermentation (Fig. 4c). Since many of these strains have been collected from fermentations (Supplementary Table 6), the substantial influence of CNVs may represent recent strong selection and adaptation to fermentation conditions that has occurred via recent CNV acquisition.

Our analysis of heritability showed that SNPs are generally able to capture most, but not all, of the genetic contribution of SVs (Fig. 4). To examine whether trait-influencing SVs would be effectively detected from SNPs alone in this population, we examined the linkage of all 113 SVs with SNPs. We found that only 63 of these SVs (55%) are in strong linkage to SNPs ($r^2 > 0.6$), leaving 45% of the SVs weakly linked. This lack of

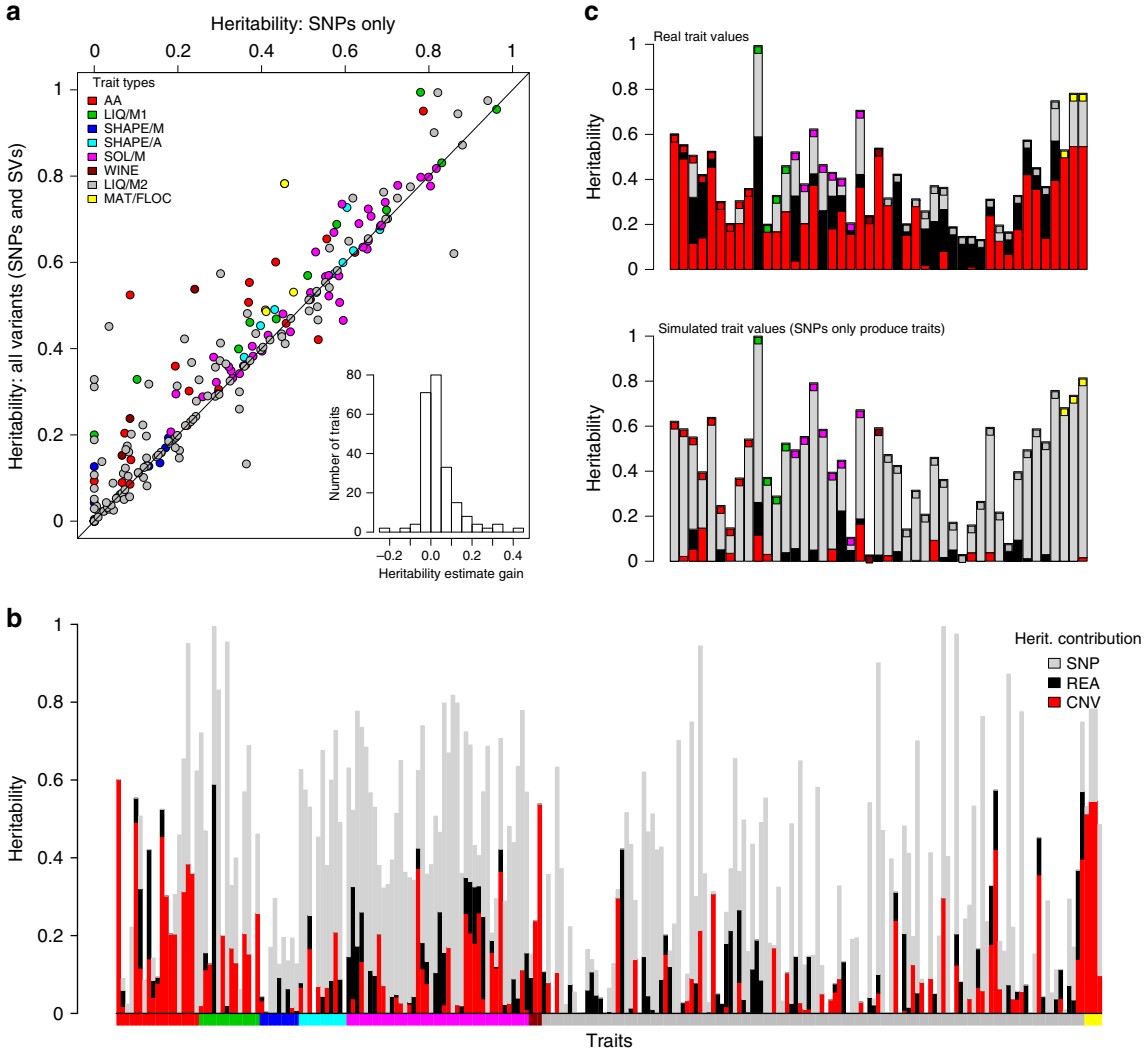

**Figure 4 | SVs contribute to quantitative traits.** (**a**) Heritability estimates are improved by the addition of SVs. Heritability estimates for 228 traits (Supplementary Table 5), using only SNP data (x axis) range from 0 to 96% (median 29%). Adding SV calls (y axis) increases the estimates (median 34%), with estimates for some traits being improved up to a gain of 43% (histogram inset). The diagonal line shows where estimates after adding SVs are the same as those without (x = y). Inset: the distribution of the 'gain' in heritability after adding SV calls (median 0.4%, maximum 43%). Points are coloured by trait types, according to legend top left. (**b**) The contributions of SNPs (grey), CNVs (red) and rearrangements (black) to heritability varied considerably between traits. Coloured bars along the x axis indicate the trait types. heritability estimates are in Supplementary Table 5. The panel below bars indicates trait types as in the legend for part (**a**). (**c**, top) For some traits, SVs explained more of the trait variation than SNPs. Boxes are coloured as legend in **a**. (**c**, lower) Analysis of simulated data generated with assumption that only SNPs cause traits indicates that the contribution of SVs to trait variance is unlikely to be due to linkage. Traits from left are; with red inset at top, free amino acid concentrations (glutamine, histidine, lysine, methionine, phenylalanine, proline and tyrosine), with green inset liquid media growth traits (maximum mass in minimal media, time to maximum slope, most rapid slope and highest cell density in rich media), in with magenta inset colony growth on solid media (with Brefeldin, CuSO$_4$, H$_2$O$_2$, hydroxyurea, 0.0025% MMS, 0.005% MMS, with proline and 0.001% SDS), wine traits with Burgundy inset (malic acid accumulation and glucose + fructose ultilisation), with grey inset liquid media conditions (caffeine lag, rate and efficiency, CsCl1$_2$ efficiency, diamide growth rate, EMS growth rate, ethanol efficiency, ethanol growth rate, galactose growth rate, growth rate at 40 °C, HqCl$_2$ lag, KCl efficiency, MgCl$_2$ efficiency, MMS lag, NiCl lag, unstressed lag and rate, SrCl efficiency, tunicamycin lag and rate), and with yellow insets mating traits (the proportion of free spores, mating figures observed and total spore counts).

linkage is consistent with SVs being transient, rather than persisting within haplotypes. Such weakly linked SVs may be missed in SNP-only association studies.

To examine this possibility, and to locate specific SVs that affect these traits, we performed mixed model genome-wide association studies, using all 68 SVs with minor allele counts ⩾5 (that is, occurring in at least five strains) as well as 139,396 SNPs and 22,058 indels with minor allele counts ⩾5. Trait-specific significance thresholds for 5% family-wise error rates were computed via permutation analysis, and

were approximately $10^{-4}$ (SVs) and $10^{-6}$ (SNPs and indels). Nineteen SVs (28%) were significantly associated with traits (15 duplications, five deletions and one translocation), as well as 228 SNPs (0.16%), and 93 indels (0.42%) (Supplementary Table 7). SVs were associated with 20 different traits, including amino acid concentrations, mating traits, and stress resistance in solid and liquid media. Nine of these SVs were not strongly linked to SNPs ($r^2 < 0.6$). The median effect size of these SVs was 14% (range 6–33%). While more detailed analyses of these associations will be required to confirm any

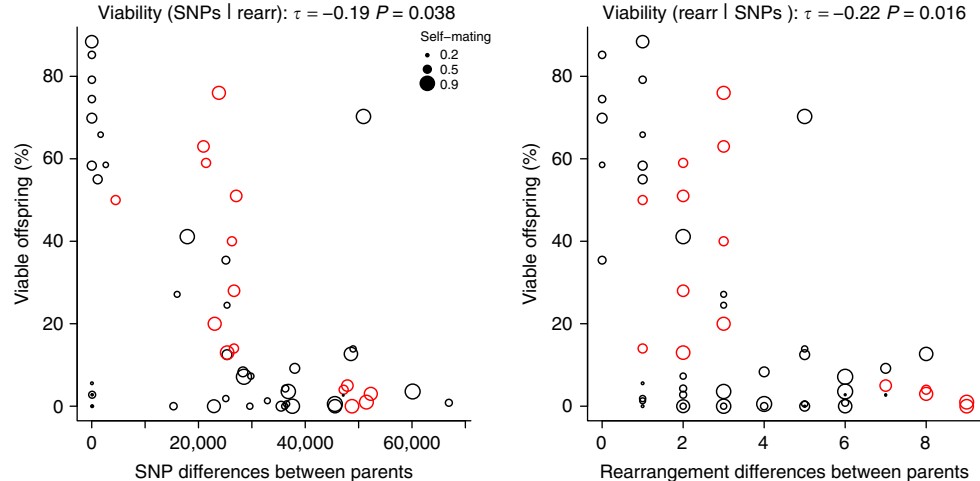

**Figure 5 | Both SNPs and rearrangements contribute to intrinsic reproductive isolation.** Spore viability was measured from 58 different crosses from Jeffares et al.[17] (black) or Avelar et al.[6] (red), with each circle in the plots representing one cross. An additive linear model incorporating both SNP and rearrangement differences showed highly significant correlations with viability ($P = 1.2 \times 10^{-6}$, $r^2 = 0.39$). Both genetic distances measured using SNPs and rearrangements (inversions and translocations) significantly correlated with viability when controlling for the other factor (Kendall partial rank order correlations with viability SNPs|rearrangements $\tau = -0.19$, $P = 0.038$; rearrangements|SNPs $\tau = -0.22$, $P = 0.016$). Some strains produce low-viability spores even when self-mated with their own genotype. The lowest self-mating viability of each strain pair is indicated by circle size (see legend, smaller circles indicate lower self-mating viability) to illustrate that low-viability outliers tend to include such cases (see Supplementary Table 8 for details).

particular association, our findings are consistent with the heritability analysis.

Collectively, these analyses indicate that even a small collection of SVs, most notably CNVs, can contribute substantially to quantitative traits. Thus, Genome-wide association studies (GWAS) analyses conducted without genotyping SVs could fail to capture these important genetic factors.

**Structural variations contribute to reproductive isolation.** Crosses between *S. pombe* strains produce between <1 and 90% viable offspring[6,18]. We have previously shown that spore viability correlates inversely with the number of SNPs between the parental strains[17]. This intrinsic reproductive isolation may be due to the accumulation of Dobzhansky–Muller incompatibilities (variants that are neutral in one population, but incompatible when combined)[33,34]. However, genetically distant strains also accumulate SVs, which are known to lower hybrid viability and drive reproductive isolation[9]. In *S. pombe*, engineered inversions and translocations reduce spore viability by ~40% (ref. 6). At present the impact of naturally occurring rearrangements, sequence divergence, and incompatible alleles in speciation within budding yeast is unclear[12–14,35,36].

To analyse intrinsic reproductive isolation in our population based on naturally occurring SVs, we examined the relationship between viability, SNPs and SVs. Both SV-distance (number of unshared SVs between parents) and SNP-distance inversely correlated with hybrid viability (Kendall correlation coefficients, SVs: $\tau = -0.26$, $P = 5.6 \times 10^{-3}$, SNPs: $\tau = -0.35$, $P = 1.6 \times 10^{-4}$) (Supplementary Fig. 9). While inversions and translocations are known to lower hybrid viability as they affect chromosome pairing and segregation during meiosis[6,18,37], CNVs are not expected to influence spore viability. Consistent with this view, there was no significant correlation between CNVs and viability (rearrangements, $\tau = -0.36$, $P = 2.0 \times 10^{-4}$; CNVs, $\tau = -0.10$, $P = 0.28$).

As the numbers of SNP and rearrangement differences between mating parents are themselves correlated ($\tau = 0.53$, $P = 1.3 \times 10^{-8}$), we also estimated the influence of each factor alone using partial correlations. When either SNPs or rearrangements were controlled for, both remained significantly correlated with offspring viability ($P = 0.04$, $P = 0.02$, respectively) (Fig. 5). Taken together, these analyses indicate that both rearrangements and SNPs contribute to reproductive isolation, but CNVs do not.

## Discussion

Here we present the first genome- and population-wide catalogue of SVs among *S. pombe* strains. To account for the high discrepancy of available methods[25], we applied a consensus approach to identify SVs (SURVIVOR), followed by rigorous filtering and manual inspection of all calls. We focused on high specificity (the correctness of the inferred SV) rather than high sensitivity (attempting to detect all SVs).

Our previous analyses of these strains, conducted without SV data[17], attributed both trait variations and reproductive isolation to SNPs and/or small indels. Here we show that the small number of SVs we describe make substantial contributions to both of these factors. We demonstrate that CNVs (duplications and deletions) contribute significantly to our ability to describe quantitative traits, whereas variants that rearrange the order of the genome (inversions and translocations) produce much weaker effects on traits. In contrast, CNVs have no detectable influence on reproductive isolation, while rearrangements contribute substantially to reproductive isolation, similar to other species[10,38].

We show that CNVs and, to a lesser extent, rearrangements can produce substantial contributions to trait variation. These CNVs subtly alter the expression of genes within and beyond the duplications, and contribute considerably to quantitative traits. Within small populations, CNVs may produce larger effects on traits in the short term than SNPs, since their effect sizes can be substantial (SVs significant in GWAS have a mean effect size of 16% in this study). Within budding yeast, clearly measured effects of alterations to gene order in the DAL metabolic cluster[39] and the lethality of some engineered rearrangements[40] indicates that rearrangements can also effect phenotypic changes. Given the evidence for extensive ploidy and

aneuploidy variation with budding yeasts, including clinical and industrial budding yeasts[29,41,42], SVs can be expected to have considerable impacts on phenotypic variation of these fungi.

In this context, it is striking that CNVs appear to be transient within the clonal populations that we studied. Our analysis is consistent with experimental studies with fission yeast, indicating that both rearrangements and CNVs may be gained or lost at rates in excess of point mutations. For example, frequent gain of duplications has been observed in laboratory cultures of *S. pombe*, where spontaneous duplications suppress *cdc2* mutants at least 100 times more frequently than point mutations. These suppressor strains lose their duplications with equal frequency[43], indicating reversion of alleles. Similarly, duplications frequently occur during experimental evolution with budding yeast[44]. This instability is likely facilitated by repeated elements, which are unstable within both budding and fission yeast genomes[45–48], which is also supported by the enrichment of SVs in our population near retrotransposon LTRs (Supplementary Fig. 10). Although we do not examine the stability of rearrangements, there is also evidence for their instability. Transposon-mediated rearrangements are highly dynamic in laboratory cultures during selection[49,50], and show elevated mutation rates at subtelomeric regions[51].

This analysis also has relevance for human diseases, since *de novo* CNV formation in the human genome occurs at a rate of approximately one CNV/10 generations[52], and CNVs are known to contribute to a wide variety of diseases[4]. Indeed, both the population genetics and the effects of SVs within *S. pombe* seem similar to human, in that CNVs are associated with stoichiometric changes on gene expression, and SVs are in weak linkage with SNPs[53,54], and therefore may be badly tagged by SNPs in GWAS studies. We show that CNVs and rearrangements in fission yeast not only rapidly emerge but also substantially contribute to quantitative traits independent of weakly linked SNPs. These findings highlight the need to identify SVs when describing traits using GWAS, and indicate that a failure to call SVs can lead to an overestimation of the impact of SNPs to traits or contribute to the problem that large proportions of the heritable component of trait variation are not discovered in GWAS (the 'missing heritability'). We observed a clear example of this effect in two winemaking traits, where heritability was entirely due to SVs.

In summary, we show that different types of SVs are transient within populations of fission yeast, where they alter gene expression, impact phenotypes and can lead to reproductive isolation.

## Methods

**Performance assessment of SV callers using simulated data.** To identify filtering parameters for DELLY, LUMPY and Pindel for the *S. pombe* genome, we simulated seven data sets (s1–s7) of $40 \times$ coverage with a range of different SV types and sizes (Supplementary Table 7). The simulated read sets contained sequencing errors (0.4%), SNPs and indels (0.1%) within the range of actual data from *S. pombe* strains and between 30 and 170 SVs. These data sets were produced by modifying the reference genome using our in-house software (SURVIVOR, described below), and simulating reads from this genome with Mason software[55].

After mapping the reads and calling SVs, we evaluated the calls. We defined a SV correctly predicted if: (i) the simulated and reported SV were of the same type (for example, duplication), (ii) were predicted to be on same chromosome and (iii) their start and stop locations were with 1 kb. We then defined caller-specific thresholds to optimize the sensitivity and false discovery rate (FDR) for each caller. FDRs on the simulated data were low: DELLY (average 0.13), LUMPY (average 0.06) and Pindel (average 0.04).

Selecting calls that were present in at least two callers further reduced the FDR (average of 0.01). DELLY had the highest sensitivity (average 0.75), followed by SURVIVOR (average 0.70), LUMPY (average 0.62) and Pindel (0.55). We further used simulated data to assess the sensitivity and FDR of our predictions. cn.mops was evaluated with a 2 kb distance for start and stop coordinates. Our cn.mops parameters were designed to identify large (above 12 kb)

events and thus did not identify any SVs simulated for s1-s6. Details of simulations and caller efficacy are provided in Supplementary Table 9.

**SURVIVOR (StructURal Variant majorIty VOte) Software Tool.** We developed the SURVIVOR tool kit for assessing SVs for short-read data that contains several modules. The first module simulates SVs given a reference genome file (fasta) and the number and size ranges for each SV (insertions, deletions, duplications, inversions and translocations). After reading in the reference genome, SURVIVOR randomly selects the locations and size of SV following the provided parameters. Subsequently, SURVIVOR alters the reference genome accordingly and prints the so altered genome. In addition, SURVIVOR provides an extended bed file to report the locations of the simulated SVs.

The second module evaluates SV calls based on a variant call format (VCF) file[56] and any known list of SVs. A SV was identified as correct if (i) they were of same type (for example, deletion); (ii) they were reported on same chromosome and (iii) the start and stop coordinates of the simulated and identified SV were within 1 kb (user definable).

The third module of SURVIVOR was used to filter and combine the calls from three VCF files. In our case, these files were the results of DELLY, LUMPY and Pindel. This module includes methods to convert the method-specific output formats to a VCF format. SVs were filtered out if they were unique to one of the three VCF files. Two SVs were defined as overlapping if they occur on the same chromosome, their start and stop coordinates were within 1 kb, and they were of the same type. In the end, SURVIVOR produced one VCF file containing the so filtered calls. SURVIVOR is available at github.com/fritzsedlazeck/SURVIVOR.

**Read mapping and detection of structural variants.** Illumina paired-end sequencing data for 161 *S. pombe* strains were collected as described in Jeffares, *et al.*[17], with the addition of Leupold's reference 975 $h^+$ (JB32) and excluding JB374 (known to be a gene-knockout version of the reference strain, see below). Leupold's 968 $h^{90}$ and Leupold's 972 $h^-$ were included as JB50 and JB22, respectively (Supplementary Table 6). For all strains, reads were mapped using NextGenMap (version 0.4.12)[57] with the following parameter (-X 1000000) to the *S. pombe* reference genome (version ASM294v2.22). Reads with 20 base pairs or more clipped were extracted using the script *split_unmapped_to_fasta.pl* included in the LUMPY package (version 0.2.9)[25] and were then mapped using YAHA (version 0.1.83)[58] to generate split-read alignments. The two mapped files were merged using Picard-tools (version 1.105) (http://broadinstitute.github.io/picard), and all strains were then down-sampled to $40 \times$ coverage using Samtools (version 0.1.18) (ref. 59).

Subsequently, DELLY (version 0.5.9, parameters: " –q 20 -r")[26], LUMPY (version 0.2.9, recommended parameter settings)[25] and Pindel (version 0.2.5a8, default parameter)[27] were used to independently identify SVs in the 161 strains using our SURVIVOR software. This included merging any variants of the same type (duplication, deletion and so on) whose start and end coordinates where within 1 kb. Merging was justified by the finding that most allele calls were close to the defined call (only 5% of start or end positions were $>300$ nt from the defined consensus boundary). We then retained all variants predicted by at least two methods. These SVs calls were genotyped using DELLY.

To identify further CNVs, we ran cn.MOPS[24] with parameters tuned to collect large duplications/deletions as follows: read counts were collected from bam alignment files (as above) with *getReadCountsFromBAM* and WL = 2000, and CNVs predicted using *haplocn.mops* with min Width = 6, all other parameters as default. Hence, the minimum variant size detected was 12 kb. CNV were predicted for each strain independently by comparing the alternative strain to the two reference strains (JB22 and JB32) and four reference-like strains that differed from the reference by $<200$ SNPs (JB1179, JB1168, JB937 and JB936).

After CNV calling, allele calling was achieved by comparing counts of coverage in 100 bp windows for the two reference strains (JB22 and JB32) to each alternate strain using custom R scripts. Alleles were called as non-reference duplications if the one-sided Wilcoxon rank sum test $P$ values for both JB22 and JB32 vs alternate strain were less than $1 \times 10^{-10}$ (showing a difference in coverage) and the ratio of alternate/reference coverage (for both JB22 and JB32) was $>1.8$ (duplications), or $<0.2$ (deletions). Manual inspection of coverage plots showed that the vast majority of the allele calls were in accordance with what we discerned by eye. These R scripts were also used to examine CNVs predicted to be segregating within clusters (clonal populations). All such CNVs were examined in all clusters that contained at least one non-reference allele call (Supplementary Table 10).

Finally, we manually mapped two large duplications that did not satisfy these criteria (DUP.I:2950001..3190000, 240 kb and DUP.I:5050001..5560000, 510 kb – both singletons in JB1207), but were clearly visible in chromosome-scale read coverage plots (Supplementary Fig. 11).

**Reduction of false discovery rate.** This filtering produced 315 variant calls. However, because 31 out of these 315 ($\sim 10\%$) were called within the two reference strains (JB22 and JB32), we expected that this set still contained false positives. To further reduce the false positive rate, we looked for parameters that would reduce calls made in reference strains (JB22 and JB32) but not reduce calls in strains more distantly related to the reference (JB1177, JB916 and JB894 that have

68223, 60087 and 67860 SNP differences to reference[17]). The reasoning was that we expected to locate few variants in the reference, and more variants in the more distantly related strains. This analysis showed that paired-end support, repeats and mapping quality were of primary value.

We therefore discarded all SVs that had a paired-end support of 10 or less. In addition, we ignored SVs that appeared in low mapping quality regions (that is, regions where reads with $MQ = 0$) or those where both start and end coordinates overlapped with previously identified retrotransposon LTRs[17].

Finally, to ensure a high specificity call set, these filtered SVs were manually curated using IGV[60] (Supplementary Tables 11 and 12). We assigned each SVs a score (0: not reliable, 1: unclear, 2: reliable based on inspection of alignments through IGV). We utilized different visualizations from IGV to identify regions were pairs of the reads mapped to different loci, for example, which we interpreted as possible artefacts. Overall, we investigated whether the alignments of the breakpoints and reads in close proximity had a reliable mapping in terms of mapping quality and clearness of the distortions of the pairs. Only calls passing this manual curation as reliable (score 2) were included in the final data set of 113 variants utilized for all further analyses. These filtering and manual curation steps reduced our variant calls substantially, from 315 to 113. At this stage only 1/113 (~1%) of these variants was called within the two standard reference strains (Leupolds's $h+$ and $h-$, JB22 and JB32 in our collection).

**PCR validation.** PCR analysis was performed to confirm 10 out of the 11 inversions and all 15 translocations from the curated data set. One inversion was too small to examine by PCR (INV.AB325691:6644..6784, 140 nt). Primers were designed using Primer3 (ref. 61) to amplify both the reference and alternate alleles. PCR was carried out with each primer set using a selection of strains that our genotype calls predict to include at least one alternate allele and at least one reference allele (usually six strains). Products were scored according to product size and presence/absence (Supplementary Tables 13 and 14).

Inversions: 9/10 variants were at least partially verified by either reference or alternate allele PCR (three variants were verified by both reference and alternate PCRs), and 7/10 inversions also received support from BLAST (see below). Translocations: 10/15 were at least partially verified by either reference or alternate allele PCR (5/15 variants were verified by both reference and alternate PCRs). One additional translocation received support from BLAST (see below), meaning that 11/15 translocations were supported by PCR and/or BLAST. Three out of the four translocations that could not be verified were probably nuclear copies of mitochondrial genes (NUMTs)[62], because one breakpoint was mapped to the mitochondrial genome. Details of the 113 curated variants are presented in Supplementary Table 15.

**Validation by BLAST of *de novo* assemblies.** We further assessed the quality of the predicted breakpoints for the inversions and translocations by comparing them to the previously created *de novo* assemblies for each of the 161 strains[17]. To this end, we created blast databases for the scaffolds of each strain that were > 1kb. We then created the predicted sequence for 1 kb around each junction of the validated 10 inversions and 15 translocations. These sequences were used to search the blast databases using BLAST+ with --gapopen 1 --gapextend 1 parameters. We accepted any blast hsp with a length > 800 bp as supporting the junction (because these must contain at least 300 bp at each side of the break point). Four inversions and three translocations gained support from these searches (Supplementary Table 2—PCR.xlsx).

**Knockout strain control.** Our sample of sequenced strains included one strain (JB374) that is known to contain deletions of the *his3* and *ura4* genes. Our variant calling and validation methods identified only two variants in this strain, both deletions that corresponded to the positions of these genes, as below:

*his3* gene location is chromosome II, 1489773-1488036, deletion detected at II:1488228-1489646.

*ura4* gene location is chromosome III, 115589-116726, deletion detected at III:115342-117145.

This strain was not included in the further analyses of the SVs.

**Microarray expression analysis.** Cells were grown in YES (Formedium, UK) and harvested at $OD_{600} = 0.5$. RNA was isolated followed by cDNA labelling[63]. Agilent $8 \times 15K$ custom-made *S. pombe* expression microarrays were used. Hybridization, normalization and subsequent washes were performed according to the manufacturer's protocols. The obtained data were scanned and extracted using GenePix and processed for quality control and normalization using in-house developed R scripts. Subsequent analysis of normalized data was performed using R. Microarray data have been submitted to ArrayExpress (accession number E-MTAB-4019). Genes were considered as induced if their expression signal after normalization was > 1.9, and repressed if < 0.51.

**Time to most recent common ancestor (TMRCA) estimates.** Previously, based on the genetic distances between these strains and the 'dated tip' dating method implemented in BEAST[64], we have estimated the divergence times between all 161 *S. pombe* strains sequenced[17]. To determine the TMRCA for pairs of strains, we re-examined the BEAST outputs using FigTree to obtain the medium and 95% confidence intervals.

**SNP and indel calling.** SNPs were called as described[17]. Insertions and deletions (indels) were called in 160 strains using stampy-mapped, indel-realigned bams as described previously[17]. We accepted indels that were called by both the Genome Analysis Toolkit HaplotypeCaller[65] and Freebayes[66], and then genotyped all these calls with Freebayes.

Briefly, indels were called on each strains bam with HaplotypeCaller, and filtered for call quality > 30 and mapping quality > 30 (bcftools filter --include 'QUAL > 30 && MQ > 30'). Separately, indels were called on each strains bam with Freebayes, and filtered for call quality > 30. All Freebayes vcf files were merged, accepting only positions called by both Freebayes and HaplotypeCaller. These indels were then genotyped with Freebayes using a merged bam (containing reads from all strains), using the --variant-input flag for Freebayes to genotyped only the union calls. Finally indels were filtered for by score, mean reference mapping quality and mean alternate mapping quality > 30 (bcftools filter --include 'QUAL > 30 && MQM > 30 & MQMR > 30'). These methods identified 32,268 indels. Only 50 of these segregated between Leupold's $h^-$ reference (JB22) and Leupold's $h^{90}$ reference (JB50), whereas 12109 indels segregated between the JB22 reference and the divergent strain JB916.

**Heredity and GWAS.** We analysed 228 traits, including those described previously[17], and three wine traits[32]. Trait values were normalized using a rank-based transformation in R, for each trait vector $y$, normal.$y$ = qnorm(rank($y$)/ $(1 + \text{length}(y))$). Total heritability, and the contribution of SNPs, CNVs and rearrangements were estimated using LDAK (version 5) (ref. 67), with kinship matrices derived from all SNPs, 146 CNVs and 15 rearrangements. All genotypes, including CNVs were encoded as binary values (1 or 0) for heritability and GWAS. To assess whether the contribution of CNVs could be primarily due to linkage with causal SNPs, we simulated trait data using the --make-phenos function of LDAK with the relatedness matrix from all SNPs, assuming that all variants contributed to the trait (--num-causals -1). We made one simulated trait data set per trait, for each of the 2 traits, with total heritability defined as predicted from the real data. We then estimated the heritability using LDAK, including the joint matrix of SNPs, CNVs and rearrangements. To assess the extent to which the contribution of SNPs to heritability was overestimated, we performed another simulation using the relatedness matrix from the 87 segregating CNVs alone, and then estimated the contribution of SNPs, CNVs and rearrangements in this simulated data as above.

Genome-wide associations were performed with LDAK using default parameters. To account for the unequal relatedness of strains, we used a kinship matrix derived from all 172,368 SNPs called previously Jeffares *et al.*[17] Association analysis was used to find associations between traits, testing SVs, SNPs and indels with a minor allele count $\geq 5$. Analysis was run separately for 68 SVs, 139,396 SNPs and 22,058 indels (each used the kinship derived from all SNPs). We examined the same 53 traits as for the heritability analysis (above). For each trait, we carried out 1,000 permutations of trait data, and define the 5th percentile of these permutations as the trait-specific $P$ value threshold.

**Model details for Heritability and GWAS Analysis.** To estimate the heritability contribution of SNPs, we computed a kinship matrix ($K_{SNP}$) using all 172,368 SNPs that we had discovered in our previous published analysis[17] (elements of this matrix represent pairwise allelic correlations across all SNPs)[67], onto which we regressed the phenotypic values assuming the following model:

$$Y \sim N(0, K_{SNP} \sigma_{SNP}^2 + \sigma_e^2 I)$$

We estimated the two variance components, $\sigma_{SNP}^2$ and $\sigma_e^2$, using REML (restricted maximum-likelihood), based on which our estimates of the heritability of SNPs is

$$\frac{\sigma_{SNP}^2}{\sigma_{SNP}^2 + \sigma_e^2}$$

To estimate the heritability of CNVs and rearrangements, we repeated this analysis using instead $K_{CNV}$ then $K_{REA}$, computed using only 146 segregating CNVs and 15 segregating rearrangements, respectively.

We additionally considered the model

$$Y \sim N(0, K_{SNP} \sigma_{SNP}^2 + K_{CNV} \sigma_{CNV}^2 + K_{REA} \sigma_{REA}^2 + \sigma_e^2 I),$$

Having estimated the four variance components, again using REML, the relative contributions of SNPs, CNVs and rearrangements are, respectively,

$$\frac{\sigma_{SNP}^2}{S}, \frac{\sigma_{CNV}^2}{S} \text{ and } \frac{\sigma_{REA}^2}{S}$$

where $S = \sigma_{SNP}^2 + \sigma_{CNV}^2 + \sigma_{REA}^2$.

To test the specificity of this analysis, we generated phenotypes for which only one predictor type contributed (for example, only SNPs), then analysed using the individual and joint models above, which allowed us to assess how accurately we can distinguish between contributions of different predictor types.

For the mixed model association analysis, we used the same the SNP kinship matrix. As the predictors (variants that we examined for effects on a trait), we chose to analyse SNPs, indels and SVs with a minor allele count $\geq 5$ (68 SVs, 139,396 SNPs and 22,058 indels).

Then for each predictor $X_j$ we considered the model

$$Y \sim N(\beta_j X_j K_{SNP} \sigma_{SNP}^2 + \sigma_e^2 I),$$

where $\beta_j$ is the effect size of predictor $X_j$

Having solved using REML, we used a likelihood ratio test (comparing to the null model ($\beta_j = 0$) to assess whether $\beta_j$ is significantly nonzero. Each of these analyses used the kinship derived from all SNPs.

**Offspring viability and genetic distance.** Cross spore viability data and self-mating viability were collected from previous analyses[6,17]. The number of differences between each pair was calculated using vcftools vcf-subset[56], and correlations were estimated using R, with the ppcor package. When calculating the number of CNVs differences between strains, we altered our criteria for 'different' variants (to merge variants whose starts and ends where within 1 kb), and merged CNVs if their overlap was >50% and their allele calls were the same.

**Transience analysis.** For each CNV, we extracted all SNPs from 20 kb upstream and 20 kb downstream. 86/87 CNVs showed variation in these regions (DUP.MT:1..19382 was the only CNV with no corresponding SNPs). We then used these concatenated SNPs to build a local SNP-based tree with FastTree (version 2.1.9) (ref. 68). To build a CNV-based tree from the copy number variation in each CNV region, we used a neighbour-joining tree estimation based on the Euclidean distances between strains.

The total branch length of the CNV-based tree was strongly correlated (Spearman rank correlation $\rho = 0.90$, $P < 0.001$) with the standard deviation of copy number variation (Supplementary Fig. 4). We therefore used this standard deviation to define a relative rate of transience for each cluster, $\sigma_{rc} = \sigma_{ic}/\sigma_{oc}$, where $\sigma_{ic}$ and $\sigma_{oc}$ are the within cluster and without cluster standard deviations, respectively, meaning that CNVs which were highly relatively transient within a given cluster would have high values of $\sigma_{rc}$. This was used to select the three CNVs visualized in Fig. 2c. See Supplementary Table 2 for all values of $\sigma_{rc}$, Supplementary Fig. 4 for visualization as heatmap. Visualizations of all 86/87 CNVs with their SNP-based phylogenies are available at: https://figshare.com/projects/fission_yeast_structural_variation/15798.

Circle plots were used to visualize the variation in copy number over the SNP-based phylogeny for each CNV using Anvi'o (version 2.0.3)[69].

**Data availability.** Sequence data are archived in the European Nucleotide Archive under study accessions PRJEB2733 and PRJEB6284. SNP, indel and SVs calls, genotypes and copy numbers are available on Figshare at: https://figshare.com/projects/fission_yeast_structural_variation/15798.

Array data is available at ArrayExpress, accession number: E-MTAB-4019.

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

## Acknowledgements

We thank Günter Klambauer for advice on cn.MOPS and Michael C. Schatz for helpful discussions and comments on the manuscript. C.J. was partly supported by a UCL Faculty of Life Science Dean s summer studentship. C.D. acknowledges support by Swiss National Science Foundation grant 150654 and UK BBSRC grant BB/M015009/1. Computations were performed on the UCL Legion and UCL Computer Science clusters. F.J.S. was supported through National Science Foundation awards (DBI-1350041) and National Institutes of Health award (R01-HG006677). D.C.J., M.H. and C.R. were supported by a Wellcome Trust Senior Investigator Award to J.B. (grant 095598/Z/11/Z). J.B. was supported by a Royal Society Wolfson Research Merit Award, L.S. is funded by an EPSRC Centre for Doctoral Training studentship at UCL CoMPLEX (EP/F500351/1).

## Author contributions

D.C.J., F.J.S., C.D. and J.B. conceived and developed the study. D.C.J., L.S., C.J. and F.J.S. conducted the bioinformatics analysis. D.C.J. designed the laboratory work. F.J.S. designed and implemented SURVIVOR. D.S. contributed to analysis of heritability and GWAS. C.R. and M.H. produced the expression array analysis. M.H. conducted PCR validation of variants. J.B. provided the bulk of funding for personnel and research costs. D.C.J., F.J.S., C.J., L.S., F.B., C.D. and J.B. wrote the manuscript.
