## [Peer Review File · Nature Communications]

Reviewers' comments:

Reviewer #1 (Remarks to the Author):

In this paper, the Authors perform a followup analysis of extent and associations of large structural variation (SV) in an *S. pombe* population genomics resource that covers 161 strains. They provide a catalog of SVs, assess their segregation patterns, correlate them to gene expression changes, and quantify linkage to cellular traits.

The paper is overall well written, and an important contribution to the understanding of extent and influences of structural variation. I only have clarifying major remarks, and a list of minor suggestions.

Major comments:

- Figure 1c and its description in caption and text are confusing:

-- Caption: "inversions often very large ... 1.04kb-5374kb". The lowest green dot is just above 0.1kb.

-- Text: "Deletions ... median length 595bp" - there are no dots below 1kb in the figure at all.

-- Text: (page 9, line 164) Median duplication size 46kb; then on page 7 line 117 the median size is 20kb.

- The copy number differences between near-clonal strains are striking and interesting.

-- You refer to these copy number changes as "transient" (subtitle) and "evolutionarily unstable" (page 10, line 191). To me, both of these imply lasting a short time, but this has not been shown.

-- Is there any way it could be due to non-chromosomal DNA content variation?

-- Did you PCR validate any sequence-inferred indel sizes?

-- It was great to read about the results in the context of genome content variation ascertained in humans (1000 genomes). It would also be relevant to compare *S. pombe* to *S. cerevisiae* and *S. paradoxus* (Bergstrom et al., 2014, MBE; especially with respect to the claims on subtelomeric gene content).

- There are gene expression changes between strains, some within duplicated regions, some outside.

The Authors ascribe these to "probably reflect indirect and compensatory effects of the ... duplication". Compensatory is also an indirect effect, making the statement tautological, but more importantly - why do you exclude the other SNPs in the strain as potential causes of the gene expression difference?

- It is not clear what went into the kinship matrices of the mixed model for each result.

-- The Methods say "composite model". P29 L576 refers to SNPs, CNVs and rearrangements included in the kinship matrix. Then on next page, 22000 indels are included as well. Where did these come from, how were they handled for heritability analyses, etc?

-- How were the SVs encoded - as a SNP for 0/1 absence/presence of the allele, or taking into account the amount of DNA that is changed as a proxy for fraction of the genome that's different?

-- Was a separate kinship matrix estimated for the SVs, and a variance component inferred for them?

-- Overall, it would be useful to have an explicit model in the Methods section from the variance parameters of which, the presented quantities are calculated. For example, what is the exact statistic on the y-axis on P42 (Fig. S8); what model does it come from, what is in the kinship matrix?

- In the discussion, the findings are framed causally (SVs make substantial contributions) - only correlations, associations, and linkages are shown throughout, so I would urge the Authors to be precise about separation of what they can show is causal from what they believe is causal.

Minor comments:

- Fig. 1a - perhaps give numbers in the pie chart as well?
- Fig. 1d - any reason to have the information of a linear genome arranged in a circle?
- Page (P) 7 line (L) 122 "Deletions and duplications and strongly biased". Also, no quantification of strength of bias in the text.
- P7 L125 - "preferentially occurred" - no quantification of preference in the text.
- P9 L174 - "significantly induced" - give amount, test statistic, p-value
- P9 L176 - "levels correlated with copy numbers" - give correlation
- P10 L178 - "no changes were evident" - give amount of change
- The information on PCR results on inversions is given in Methods; I would not have expected to see novel results there.
- P12 L219 "CNVs influence quantitative traits" - only linkage is shown, no causality.
- P19 L366 "measurable rate" - all such rates are measurable :)

Reviewer #2 (Remarks to the Author):

The authors present an analysis of structural variants in their recently sequenced *S. pombe* natural isolates. They show structural variation, including what must be segregating structural variation, contributes to changes in gene expression, substantially to quantitative trait variation, and potentially to reproductive isolation. Overall the analyses seem robust and the manuscript is well written and easy to follow.

The study contributes to the establishment of *S. pombe* as a model of natural variation and quantitative traits. My one criticism would be that similar things have been shown previously in both human and *S. cerevisiae* quantitative trait analyses. The authors need to better distinguish their findings from what has previously been shown in these other more intensively studied species.

Reviewer #3 (Remarks to the Author):

In this manuscript, Jeffares et al. analyze published short read data to detect structural variants (SVs) in the genomes of a set of *S. pombe* strains. They associate these SVs with phenotypic variation among the strains and provide estimates of heritability explained by SVs, as well as map individual cases of SV-trait associations. Perhaps the most interesting and surprising result is that nearly clonal groups of strains that are almost identical at the SNP level nevertheless segregate for several SVs. The role of SVs in complex trait variation is interesting, and this is an interesting contribution. The manuscript is clearly written, follows a clean logic, and therefore is easy to follow.

I would like to see additional information on some of the results, mainly about the SVs that segregate within clonal populations. I also have suggestions for minor additions and clarifications.

Main comments:

1. The most surprising result is that the nearly clonal sets of strains do segregate SVs. I would like to see a little more information about these events in the main text. Specifically:
 - a) What are the allele frequencies within the clonal populations? Is it usually the case that only a single member of a clonal set carries the SV, or are they at higher frequency?
 - b) Within the clonal populations, do the SV alleles reflect relatedness based on SNPs? I realize this analysis may be underpowered because of the low number of SNPs, but maybe there are obvious agreements between SV status and SNP alleles, perhaps splitting each clonal group in half? Or do the

SVs just occur randomly within each cluster?

c) Is there segregation of CNV copy number within each cluster of clonal strains? I.e., do all clonal strains that carry a given CNV have the same copy number, or is there variation in how many copies they carry?

2. p. 8 l. 147 "Furthermore, we observed instances of the same SVs that were present in two or more different clonal populations that were not fixed within any clonal population." This is a really interesting observation, and I would like to hear more about it in the Results and / or the Discussion. Currently there is only this one sentence for a topic that could easily support an entire paragraph:

a) Are these SVs shared perfectly (with the same breakpoints) or do they just overlap?

b) Are there any patterns in terms of which clonal populations share a given SV? For example, are the clonal populations that share a given SV more closely related to each other than clonal populations that do not share SVs?

c) Do you see any evidence that these SVs might have been moved around between populations by outcrossing?

d) If not, what is your explanation for why these SVs occur? Are they recurrent mutations at labile sites in the genome that are more prone to forming SVs?

e) If they are recurrent mutations, can you infer or speculate about the mutational mechanism? For example, are the SV break points close to repetitive elements or close gene paralogs that might frequently create errors in recombination?

f) For shared CNVs, is the copy number of the CNV the same or different in different clonal populations?

3. Do any of the SVs that are associated with a phenotype segregate within a clonal population? If yes, how much of trait variance does the SV explain within that clonal population? Because there is essentially no other genetic variation among the clones, the SV might completely determine genetic trait variation among the clones. It would be interesting to know if such cases exist.

4. Please provide a supplementary text or spreadsheet file that lists the genotypes (presence / absence and copy number where appropriate) for each SV in each strain. This would also help address some of my questions above on allele frequencies and SV sharing. Together with the phenotypes that are available from reference 8, this would allow readers to recapitulate the heritability and association analyses. I couldn't find SNP genotypes associated with reference 8 (although I checked only briefly). These would also need to be made available to ensure that readers can reproduce the analyses presented here. If they are available somewhere, a brief mention of their location would be useful in the present paper.

5. Have you done qPCR to confirm some of the CNVs, especially those that segregate among multiple clonal populations?

6. In the visual inspections for the SV calls, what types of artifacts or features did you look for? What were typical failure modes for putative SVs that you deemed incorrect? A brief description in the Methods would be useful to the community.

Minor comments:

7. Supplementary Figure S3: It would be helpful to indicate the absolute coverage of the strains as well. This would help to get a better sense of the strength of the signal. For example, a two-fold coverage difference means more with a 100X coverage baseline than a 2X baseline. If different strains had different average genome coverage, how were the relative coverages in the plots calculated?

Were they anchored to the flanking sequence somehow, or are they purely "coverage strain 1 / coverage strain 2"? I'm trying to understand why some of the green strains in the figure have less coverage than the reference. The normalization scheme would probably explain this.

8. p. 14 l. 262 "Our analysis of heritability showed that SNPs are generally able to capture most of the genetic contribution of SVs" seems to contradict the result on p. 13 l. 233 that "Analysis of simulated data confirmed that the contribution of CNVs could not be explained by linkage to causal SNPs alone". Please clarify.

9. p. 17 l. 311 "we found that rearrangements explained spore viability better than CNVs [...]" this implies that you tested rearrangements and CNVs directly against each other, perhaps as you did further down for SNPs and rearrangements. Please rephrase this to "while rearrangements correlated with spore viability, there was no significant correlation between CNVs and viability".

10. Figure 4: the legend has an incorrectly rounded p-value: SNPs | rearrangements = 0.03, whereas the figure gives $p = 0.038$, which is $p = 0.04$ after rounding. The correlation estimate is also slightly different between legend and figure.

11. p. 19 l. 366 Instead of a "measureable" rate, do you mean "considerable" or simply "high"? All mutation rates can be measured.

12. Supplemental Figure S8: in the top left panel, in the leftmost bar, the open circle above the bar should probably be filled? If not, why is the "estimate - 1sd" higher than the estimate?

13. Abstract: "genomics regions" should be "genomic regions"

Reviewer #4 (Remarks to the Author):

This paper focuses on the effects of structural variation on phenotypic differences and reproductive isolation in *Sc. pombe*. Although the work in this manuscript is performed well, I had some significant criticisms:

1. Not very much was done with the ample phenotype dataset to make specific connections between genetic variants and traits.

2. It is known that *Sc. pombe* isolates exhibit a substantial amount of structural variation. This paper improves upon our knowledge of this details of this structural variation, but at this juncture, these details seem to represent an incremental advance.

3. A large amount of work in *Sa. cerevisiae* has shown that structural variation can have important phenotypic and gene expression effects, and that some of these structural variations can be transient. I thought the attempt to determine the quantitative contribution of structural variation to phenotypic variation was of value, but the insights gained also seemed incremental.

4. A number of papers in *Saccharomyces* have shown the transient nature of structural variation.

5. Extensive work by Gianni Liti and Ed Louis on reproductive isolation in *Saccharomyces*, especially *Sa. paradoxus*, already has shown relationships between amount of structural variation and

reproductive isolation. The fact that CNVs may not impact this relationship is to be expected.

6. The aesthetics of the figures could be improved; e.g., Fig 3 might be better if plots with points instead of bars were used and Fig 4 might be aided by a legend panel indicating the difference between red and black points or differently sized points.

7. It was surprising that more work from *Saccharomyces* was not cited. This was especially true in the section on reproductive isolation, where the work mentioned above, which arguably represents the gold standard for yeast papers on the topic, was not even recognized. Ultimately, many of the questions addressed in this paper have been extensively examined in *Saccharomyces*. Even though this is a different yeast genus, it is still important to cite and discuss the prior work in *Saccharomyces* and describe how this paper builds upon it.

In summary, the science and writing in this paper were solid. However, this paper had insufficient novelty and awareness of historical context to warrant publication in *Nature Communications*.

REVIEWERS' COMMENTS:

Reviewer #1 (Remarks to the Author):

The Authors have thoroughly addressed my previous comments, and I have no further ones to make. I defer to other Reviewers in regard to novelty of the findings in *S.pombe*, as this is not my area of expertise.

Reviewer #2 (Remarks to the Author):

I am satisfied with the response to my comments - the text changes now better distinguish this study from previous work in the other yeast + human populations.

Reviewer #3 (Remarks to the Author):

Thanks to the authors for carefully addressing all my previous comments. I have just a few minor remaining comments:

1. p. 9 l. 160 refers to Figure 3c, which does not seem right. Should this be Figure 2c or some other Figure?
2. p. 9 l. 165: "strong correlation between the total mutation in these regions and the total variation in copy number of the CNV" is awkwardly phrased. "total mutation" sounds like it includes the CNV, which seems wrong. Please reword.
3. Supplementary Figure 4 a & b share one axis, but in a) it is the x-axis while in b) it is on the y axis. Please make this consistent.
4. There are two each of Supplementary Figures 6 and 7. Please fix.
5. In the second Supplementary Figure 7, the top middle and top right panels have the same axis labels, but show different data. Please clarify.

Reviewer #4 (Remarks to the Author):

The revised version of this manuscript represents a significant improvement over the initial submission. The authors do a much better job now of connecting their work to previous papers from other groups, including labs that work on *Saccharomyces cerevisiae*. It is clearer how the components of the paper collectively build into a manuscript that could be of value to a number of different groups of researchers (e.g., people working on *S. pombe*, quantitative genetics, folks interested in structural variation).

Aside from one comment, I am satisfied with how the authors addressed my remarks and handled input from the other reviewers. However, there was a misinterpretation of my first point, perhaps because I could have been clearer: 'Not very much was done with the ample phenotype dataset to make specific connections between genetic variants and traits'. What was meant by this point is that the authors do not discuss how any specific variants influence any specific traits? In other words, no

discussion of the molecular and systems mechanisms contributing to heritable phenotypic variation in this organism is provided. For example, on p15, the authors write: 'Thus, some groups of traits have consistently larger contributions from SVs than from SNPs alone. These traits include intracellular amino acid concentrations...' Can you make any connection to the mechanisms based on SNVs? This seems especially feasible for CNVs, which are often resolved to individual genes. There are other similar opportunities in the paper. I don't think these modifications are absolutely necessary, but they would certainly help make this paper more accessible to researchers who are not statistical geneticists.

More minor comments:

Yeast species: Often in a context where both *Saccharomyces cerevisiae* (or related species) and *Schizosaccharomyces pombe* are being discussed, the former and latter will be referred to as *Sa. cerevisiae* and *Sc. pombe*, respectively, to prevent confusion.

P3, l68: The word 'progress' read weird to me. Maybe 'advance'?

P4, l71: 'Various aspects of biology' is a vague phrase.

P21, l398-402: In these sentences the authors mention experimental studies in budding yeast, but then an example from *S. pombe* is provided.

Reviewer #1

In this paper, the Authors perform a follow up analysis of extent and associations of large structural variation (SV) in an *S. pombe* population genomics resource that covers 161 strains. They provide a catalog of SVs, assess their segregation patterns, correlate them to gene expression changes, and quantify linkage to cellular traits. The paper is overall well written, and an important contribution to the understanding of extent and influences of structural variation. I only have clarifying major remarks, and a list of minor suggestions.

Major comments:

1.1) Figure 1c and its description in caption and text are confusing:

1.1.1) Caption: "inversions often very large ... 1.04kb-5374kb". The lowest green dot is just above 0.1kb.

1.1.1) Text: "Deletions ... median length 595bp" - there are no dots below 1kb in the figure at all.

1.1.1) Text: (page 9, line 164) Median duplication size 46kb; then on page 7 line 117 the median size is 20kb.

Response: *Yes, some of these numbers were incorrect. Thanks for pointing this out. We have adjusted the text so that it correctly represents our findings:*

"inversions often very large, spanning large portions of chromosomes (0.1 kb–5,374 kb"
and

*"The deletions were generally smaller (median length 14 kb, **Figure 1c**), and duplications slightly larger (median length of 21 kb),"*

and

"The naturally occurring duplications we described are typically smaller (median length: 21 kb), including an average of 6.5 genes."

1.2) The copy number differences between near-clonal strains are striking and interesting.

Response: *We agree and thank the reviewer for his/her endorsement.*

1.2.1) You refer to these copy number changes as "transient" (subtitle) and "evolutionarily unstable" (page 10, line 191). To me, both of these imply lasting a short time, but this has not been shown.

Response:

We have performed additional, quantitative investigations of the transience of CNVs, and report these in a new section with new figures. In brief, we constructed local, SNP-based phylogenies for the region surrounding each CNV (20kb up- and down-stream, merged) and found that strains identical in these regions could still have different copy numbers within clusters of near-clonal strains (Figure 2). This high similarity (as well as the near-identical sequence throughout the genome for clonal clusters) effectively rules out CNV gain/loss by recombination. We also produced neighbor joining trees from CNV alleles, and showed that CNV-allele distance, and local SNP-tree distance were only weakly correlated, consistent with different processes. We also extended the discussion, relating our analysis to the previously published information from analysis of budding yeast populations, and laboratory work in budding & fission yeast showing that repetitive elements are unstable (have high mutations rates). All these analyses and considerations consistently support the notion that copy number variants (and perhaps inversions and translocations) are often transient within populations.

1.2.2) Is there any way it could be due to non-chromosomal DNA content variation?

Response: *To us, this seems unlikely. Because the DNA samples that was sequenced is derived from a single colony that is then used to start a 50 mL overnight culture. Since this involves many cell divisions, any non-chromosomal DNA would need to contain origins of replication and centromeric elements to be properly segregated during this growth.*

1.2.3) Did you PCR validate any sequence-inferred indel sizes?

Response: *No, we did not, because large deletions and duplications are clearly visible by read coverage. Insertions were not robustly inferred by the methods (Delly, Lumpy, Pindel). We focused our PCR on validating breakpoints for inversions and translocations. However the expression arrays we performed effectively validated 7 of the 8 duplications*

that were not shared within a cluster (because they all showed an increase in expression level – an unlikely occurrence without duplication). By ‘indels’ here, we assume that the reviewer is referring to large deletions and/or duplications – small indels were detected using HaplotypeCaller and Freebayes (see methods).

1.2.4) It was great to read about the results in the context of genome content variation ascertained in humans (1000 genomes). It would also be relevant to compare *S. pombe* to *S. cerevisiae* and *S. paradoxus* (Bergstrom et al., 2014, MBE; especially with respect to the claims on subtelomeric gene content).

Response: *Yes, the Bergstrom paper¹ produces some finding that similar to ours. That CNVs and rearrangements are more abundant in subtelomeric regions, that a CNV appears to associate with a quantitative trait (arsenic resistance in this case), and that subtelomeric regions tend to have more loss-of-function variants (we observed in a similar thing, along with an increase of retrotransposons in our Nature Genetics paper²). They also note that the strains that have SV similarity tend to have high spore viability, consistent with our findings (but do not show any correlation, or statistics). We have highlighted these similarities in the manuscript.*

1.3) There are gene expression changes between strains, some within duplicated regions, some outside. The Authors ascribe these to "probably reflect indirect and compensatory effects of the ... duplication". Compensatory is also an indirect effect, making the statement tautological, but more importantly - why do you exclude the other SNPs in the strain as potential causes of the gene expression difference?

Response: *Yes, there is a possibility that SNPs or indels might cause changes in expression levels, particularly outside of the duplications. We have modified our text to express the more circumspect view that expression changes outside of duplications might be due to other variants (we also modified figures, tables & text with these numbers). That being said, the main conclusions are still valid: First, that we could validate the genomic duplications that we predicted. Second, that duplications cause unbuffered changes in gene expression, and that these duplications*

segregate between very closely related strains. The text about gene expression changes outside of duplications now reads:

“As environmental growth conditions were tightly controlled, these changes in gene expression could be due to either compensatory effects of the initial perturbation caused by the duplication or changes that arise due to SNPs or indels that segregate between the strains”.

1.4) It is not clear what went into the kinship matrices of the mixed model for each result.

1.4.1) The Methods say "composite model". P29 L576 refers to SNPs, CNVs and rearrangements included in the kinship matrix. Then on next page, 22000 indels are included as well. Where did these come from, how were they handled for heritability analyses, etc?

Response: *For both the heritability analysis and the GWAS we used a kinship matrix generated from all SNPs – we did not use indels for kinship. We used the 22,058 indels as predictors (variants that may affect traits). We have adjusted the relevant section of the Methods section to clarify our procedure, as follows:*

To estimate the heritability contribution of SNPs, we computed a kinship matrix (K_{SNP}) using all 172,368 SNPs that we had discovered in our previous published analysis² (elements of this matrix represent pairwise allelic correlations across all SNPs)³, on to which we regressed the phenotypic values assuming the following model:

$$Y \sim N(0, K_{SNP} \sigma_{SNP}^2 + \sigma_e^2 I)$$

We estimated the two variance components, σ_{SNP}^2 and σ_e^2 , using REML (restricted maximum likelihood), based on which our estimates of the heritability of SNPs is

$$\frac{\sigma_{SNP}^2}{\sigma_{SNP}^2 + \sigma_e^2}$$

To estimate the heritability of CNVs and rearrangements, we repeated this analysis using instead K_{CNV} then K_{REA} , computed using only 146 segregating CNVs and 15 segregating rearrangements, respectively.

We additionally considered the model

$$Y \sim N(0, K_{SNP}\sigma_{SNP}^2 + K_{CNV}\sigma_{CNV}^2 + K_{REA}\sigma_{REA}^2 + \sigma_e^2 I),$$

Having estimated the four variance components, again using REML, the relative contributions of SNPs, CNVs and Rearrangements are, respectively,

$$\frac{\sigma_{SNP}^2}{S}, \frac{\sigma_{CNV}^2}{S} \text{ and } \frac{\sigma_{REA}^2}{S}$$

$$\text{where } S = \sigma_{SNP}^2 + \sigma_{CNV}^2 + \sigma_{REA}^2$$

To test the specificity of this analysis, we generated phenotypes for which only one predictor type contributed (e.g., only SNPs), then analyzed using the individual and joint models above, which allowed us to assess how accurately we can distinguish between contributions of different predictor types.

For the mixed model association analysis, we used the same the SNP kinship matrix. As the predictors (variants that we examine for effects on a trait), we chose to analysis SNPs, indels and SVs with a minor allele count ≥ 5 (68 SVs, 139396 SNPs and 22,058 indels).

Then for each predictor X_j we considered the model

$$Y \sim N(\beta_j X_j K_{SNP}\sigma_{SNP}^2 + \sigma_e^2 I),$$

where β_j is the effect size of predictor X_j

Having solved using REML, we used a likelihood ratio test (comparing to the null model ($\beta_j = 0$)) to assess whether β_j is significantly non-zero. Each of these analysis used the kinship derived from all SNPs.

1.4.2) How were the SVs encoded - as a SNP for 0/1 absence/presence of the allele, or taking into account the amount of DNA that is changed as a proxy for fraction of the genome that's different?

Response: *All variants were encoded as 1 or 0 (haploid) for heritability and GWAS analysis. We have added this sentence to the methods "All genotypes, including copy number variants were encoded as binary values (1 or 0) for heritability and GWAS."*

1.4.3) Was a separate kinship matrix estimated for the SVs, and a variance component inferred for them?

Response: *Yes, as described above, we computed one kinship matrix for CNVs and one for rearrangements. The all SNP, CNV and rearrangement kinship matrices were used to estimate the relative contributions of SNPs, CNVs and Rearrangements to heritability under a joint model. For GWAS, only the all SNP kinship was used, because this accounts for the null model that most variants do not make significant contributions to the trait.*

1.4.4) Overall, it would be useful to have an explicit model in the Methods section from the variance parameters of which, the presented quantities are calculated. For example, what is the exact statistic on the y-axis on P42 (Fig. S8); what model does it come from, what is in the kinship matrix?

Response: *We have defined a more explicit model, which we now include in the methods.*

1.5) In the discussion, the findings are framed causally (SVs make substantial contributions) - only correlations, associations, and linkages are shown throughout, so I would urge the Authors to be precise about separation of what they can show is causal from what they believe is causal.

Response: *There are always uncertainties in play when interpreting high throughput studies, genomics and quantitative genetics analysis, and so we tried to be conservative and circumspect about our analysis. This is particularly so for GWAS, where causal variants are much more difficult to prove than they are to infer statistically.*

Our aim was to describe the effects of SVs on a genome-wide scale, over many traits. Given the quality of data, we chose to describe the whole analysis, rather than to show that any particular variant was causal.

We believe that our methods are sufficiently rigorous to support the conclusions. We describe tests and highlight p-values for all results, when these were significant – or interestingly not so. E.g. that copy number variations are not correlated with spore viability, whereas rearrangement differences were significantly correlated. We also

corrected for biases, where we could, such as controlling for linkage effects when estimating heritability (using simulated data), and using partial correlations to control for the correlation between SNP parental distance and SV-parental distance in spore viability analysis. All details are in the results or in the supplementary tables.

Minor comments:

- Fig. 1a - perhaps give numbers in the pie chart as well?

Response: *We have done this.*

- Fig. 1d - any reason to have the information of a linear genome arranged in a circle?

Response: *We think that is a useful way of showing inversions and translocations. We show the same information in a linear way in supplementary Figure 1. If the reviewers or editors like this better we are happy to swap Fig. 1d and Supp. Fig 1.*

- Page (P) 7 line (L) 122 "Deletions and duplications and strongly biased". Also, no quantification of strength of bias in the text.

- P7 L125 - "preferentially occurred" - no quantification of preference in the text.

Response: *Quantifications and plots are shown in Supplementary Figure 2, with P-values and methods, where they don't disrupt the flow of the paragraph. Since the bias for rearrangements is slight we dropped the word strongly from the sentence in the main text. We now refer to Supplementary Figure at this point in the main text. We adjusted this legend to be a little more descriptive, to read: "Both CNVs and rearrangements are biased towards the ends of chromosomes (CNVs; median distance to chromosome ends 236 kb vs chromosome- and size matched random sites 944 kb, Wilcoxon rank sum test $P = 1.3 \times 10^{-11}$, rearrangements median distance 569 kb vs matched random 863 kb, Wilcoxon test $P = 0.03$)."*

- P9 L174 - "significantly induced" - give amount, test statistic, p-value

Response: *P-values in Fig 2c legend.*

- P9 L176 - "levels correlated with copy numbers" - give correlation

Response: *Spearman rank correlation $\rho = 0.71$ and $P = 0.014$ now in text.*

- P10 L178 - "no changes were evident" - give amount of change

Response: *Relative gene expression changes within the 50kb region adjacent to each duplication are shown in Supplementary Figure 4 (which we refer to), with P-values in the legend. Though some are nominally significant (P values 0.04 and 0.03, these wouldn't survive Bonferroni's correction, and they are not even nominally significant if we use a 40kb window).*

- The information on PCR results on inversions is given in Methods; I would not have expected to see novel results there.

Response: *We prefer not to place such detailed results in the main text. We provide a summary in the Methods, and detailed results in Supplementary Tables 14 and 15.*

-P12 L219 "CNVs influence quantitative traits" - only linkage is shown, no causality.

Response: *With this kind of population data (genotypes, trait values) it is simpler to show that a particular set of variants contribute to a trait using heritability analysis, than to show a statistically significant effect of a single variant with GWAS (the closest to causal we'll get with genome-wide analysis). So we analyze heritability first. What we do show clearly via careful analysis accompanied by simulations that control for linkage of alleles, is that Copy number variants contribute to the heritability of quantitative traits. We have altered this subheading to: "Copy number variants contribute to the heritability of quantitative traits".*

- P19 L366 "measurable rate" - all such rates are measurable :)

Response: *We altered this sentence to: "at a rate of approximately one CNV/10 generations".*

Reviewer #2

The authors present an analysis of structural variants in their recently sequenced *S. pombe* natural isolates. They show structural variation, including what must be segregating structural variation, contributes to changes in gene expression, substantially to quantitative trait variation, and potentially to reproductive isolation. Overall the analyses seem robust and the manuscript is well written and easy to follow.

2.1) The study contributes to the establishment of *S. pombe* as a model of natural variation and quantitative traits. My one criticism would be that similar things have been shown previously in both human and *S. cerevisiae* quantitative trait analyses. The authors need to better distinguish their findings from what has previously been shown in these other more intensively studied species.

Response: *Yes, somewhat related analyses have been done in budding yeast¹, and hints of similar phenomena are present in human/animal GWAS & structural variant literature. We now cite several additional budding yeast papers, owing to the encouragement of another reviewer. Our analysis has seven novel features that are not present in any one paper, or analyzed with one population:*

- 1. We conduct a genome and population-wide screen of both rearrangements and copy number variants (there is no such screen for budding yeast).*
- 2. We quantitatively analyze their impact on spore viability, taking SNP correlations into account (not done for any species as far as we know).*
- 3. We conduct heritability analysis using SNPs, both rearrangements and copy number variants (again no budding yeast analysis, nor human).*
- 4. We conduct GWAS analysis using SNPs, both rearrangements and CNVs.*
- 5. We quantitatively describe the transient nature of CNVs in micro populations (hinted at in human genome paper).*
- 6. We show that rearrangements and copy number variants have differences in their effects on heritability/traits and reproductive isolation.*
- 7. We introduce and release a novel software tool for consensus calling of SVs ("SURVIVOR"), which further includes a simulation and evaluation framework for all types of SVs.*

Reviewer #3

In this manuscript, Jeffares et al. analyze published short read data to detect structural variants (SVs) in the genomes of a set of *S. pombe* strains. They associate these SVs with phenotypic variation among the strains and provide estimates of heritability explained by SVs, as well as map individual cases of SV-trait associations. Perhaps the most interesting and surprising result is that nearly clonal groups of strains that are almost identical at the SNP level nevertheless segregate for several SVs. The role of SVs in complex trait variation is interesting, and this is an interesting contribution. The manuscript is clearly written, follows a clean logic, and therefore is easy to follow. I would like to see additional information on some of the results, mainly about the SVs that segregate within clonal populations. I also have suggestions for minor additions and clarifications.

Response: *We thank the reviewer for their interest and endorsement. We have produced a new quantitative analysis of the transient nature of CNVs, including some figures.*

Main comments:

3.1 The most surprising result is that the nearly clonal sets of strains do segregate SVs. I would like to see a little more information about these events in the main text.

Specifically:

3.1.1) What are the allele frequencies within the clonal populations? Is it usually the case that only a single member of a clonal set carries the SV, or are they at higher frequency?

Response: *Within clusters, non-reference allele counts were low (median 1, mean 2.1), consistent with variants that are transient (appear relatively frequently, but do not remain in small populations). This pattern was observed with both duplications and deletions (where the non-reference allele is most likely the derived allele). With inversions/translocations (where its more difficult to tell derived/ancestral alleles) minor allele frequencies within cluster are low. We added this in the text, “Finally, we found that many CNVs represented the rare allele within the cluster, consistent with events that have short half-lives (Supplementary Figure 5).”, and we provide a supplementary figure.*

3.1.2) Within the clonal populations, do the SV alleles reflect relatedness based on SNPs? I realize this analysis may be underpowered because of the low number of SNPs, but maybe there are obvious agreements between SV status and SNP alleles, perhaps splitting each clonal group in half? Or do the SVs just occur randomly within each cluster?

Response: *Thank you for pointing this out. We have performed additional analyses examining this. It appears that there is no such obvious agreement: Because different regions of the genome share different histories, we constructed SNP-based phylogenies using the 20kb up- and down-stream regions (merged) of each CNV, which gave an average of ~140 SNPs for each CNV. There was only weak correlation between the total branch length of a SNP-based maximum-likelihood tree compared to a copy-number-based neighbor-joining tree (Figure 2a) suggesting that CNVs are subject to additional or different evolutionary processes and that their variation cannot be explained by relatedness based on SNPs. Looking within clusters, some CNVs showed high relative rates of variation in copy number compared to the rest of the dataset (Figure 2b). Additionally, the SNP-based phylogenies for these CNVs could not resolve closely-related strains within clusters that had different copy numbers (Figure 2c).*

See also response to 1.2.1

3.1.3) Is there segregation of CNV copy number within each cluster of clonal strains? I.e., do all clonal strains that carry a given CNV have the same copy number, or is there variation in how many copies they carry?

Response: *Yes, when there is a duplication within a cluster, we do see some copy number variations within the cluster. We show some examples in Supplementary Figure 3. We show in our new analysis of copy number variance vs SNP diversity that some duplications have high variance.*

3.2) 2. p. 8 l. 147 "Furthermore, we observed instances of the same SVs that were present in two or more different clonal populations that were not fixed within any clonal population." This is a really interesting observation, and I would like to hear more about it in the Results and / or the Discussion. Currently there is only this one sentence for a

topic that could easily support an entire paragraph:

Response: *We agree that this is a key observation of our study, and the revised manuscript analyses and discusses this phenomenon in more depth. We produced a new analysis of the transient nature of CNVs, which is consistent with the variants being transient. We mention the previous report of the loss of a duplication in laboratory conditions⁴. We also discuss the fact that duplications contain a repeat of a sequence, and it is widely known that repeated elements are unstable⁵⁻⁸. See page 9 of the manuscript “To examine whether this transience is a general feature of CNVs in this population, we quantified ...”, and Figure 2.*

3.2.1) Are these SVs shared perfectly (with the same breakpoints) or do they just overlap?

Response: *There were 15 non-overlapping duplications that were present in two or more different clonal populations that were not fixed within any. Only three that didn't overlap at all (as seen from below).*

Number clusters segregating	Variant (type.chr:start..end)
2	DUP.I:5448001..5460000
5	DUP.I:5542001..5562000
11	DUP.I:5544001..5562000
12	DUP.I:5546001..5562000
5	DUP.II:2116001..2134000
4	DUP.III:212001..258000
3	DUP.III:214001..286000
2	DUP.III:220001..300000
2	DUP.III:222001..296000
5	DUP.III:224001..264000
6	DUP.III:234001..254000

6	DUP.III:236001..256000
3	DUP.III:246001..278000
3	DUP.III:274001..286000
4	DUP.III:275493..284754

We looked at two of these. One appears to share the same breakpoints between clusters (top plot below), while the other does not (lower plot). Overall, though, the small number of events makes it difficult to draw general conclusions.

3.2.2) Are there any patterns in terms of which clonal populations share a given SV? For example, are the clonal populations that share a given SV more closely related to each other than clonal populations that do not share SVs?

Response: *This is a good question. It is dealt with by in our new analysis of the transience of CNVs. The genetic distance between SVs and SNPs around in the SV are only weakly correlated, consistent with some allele sharing. The existence of (weak) LD between SNPs and SVs also suggests between-cluster allele sharing. But because CNVs can vary within clusters, and there is evidence that they can revert, we know that SNP-CNV relatedness will be relatively low.*

3.2.3) Do you see any evidence that these SVs might have been moved around between populations by outcrossing?

Response: *Yes. We have shown previously that these strains have had recombination between them². So SV sharing by outcrossing would be the simple explanation for any*

that has a minor allele > 1 within the non-clonal 57 strains (ie: outside of a cluster). We examined this: for the 86 SVs that are present in our selected set of 57 non-clonal strains, 61 are present in >1 strain.

3.2.4) If not, what is your explanation for why these SVs occur? Are they recurrent mutations at labile sites in the genome that are more prone to forming SVs?

Response: *Outcrossing would be the simplest explanation for those where the SNP-tree distance (from the SNPs round the SV) is approximately equal to the CNV-tree distance. In those cases where it is not, there are two possible scenarios: independent gain in >1 lineage, or independent loss. We favor independent loss for duplications, because it is easy to imagine how a homologous repair might condense a two copy duplicate back down to one copy, and relatively frequent loss has been shown to occur in laboratory conditions⁴. Similarly, rearrangements (inversions/translocations) are known to be facilitated by repetitive elements (Ty in budding yeast), so it is plausible that they might reoccur. Unfortunately the genetic distances within our clonal clusters did not allow us to model SV birth and death.*

3.2.5) If they are recurrent mutations, can you infer or speculate about the mutational mechanism? For example, are the SV break points close to repetitive elements or close gene paralogs that might frequently create errors in recombination?

Response: *We found that all classes of SVs were enriched close to LTR elements of the Tf1-type retrotransposons that are present in *S. pombe* strains. We have added a sentence to the discussion, “This instability is likely facilitated by repeated elements, which are unstable within both budding and fission yeast genomes⁴⁵⁻⁴⁸, which is also supported by the enrichment of SVs in our population near retrotransposon LTRs (Supplementary Figure 8).”.*

3.2.6) For shared CNVs, is the copy number of the CNV the same or different in different clonal populations?

Response: *The copy number of the CNV varies within and between clusters. We provide one example in Figure 3, and the remainder as supplementary figures.*

3.3). Do any of the SVs that are associated with a phenotype segregate within a clonal population? If yes, how much of trait variance does the SV explain within that clonal population? Because there is essentially no other genetic variation among the clones, the SV might completely determine genetic trait variation among the clones. It would be interesting to know if such cases exist.

Response: *Yes, there were 10 SVs that were associated with a trait, and also segregated within at least one cluster. Since the estimated effect sizes range from 6% to 25%, and there are < 150 SNPs within clusters, we would expect the SV to be the major variant affecting within-cluster phenotypic diversity in some cases. However, we did not do this analysis, because of the large number of tests required; many traits, and many clusters, and the low power we would achieve. To satisfy our curiosity (and hopefully your own) we did one analysis: to estimate the contribution of SNPs and SVs to heritability in one cluster04, which has 8 strains. In this analysis, for the 45 traits with some variation, 35% (16/45) had a greater contribution from SVs than SNPs, confirming the expectation that SVs will be the major variants involved for some traits.*

3.4). Please provide a supplementary text or spreadsheet file that lists the genotypes (presence / absence and copy number where appropriate) for each SV in each strain. This would also help address some of my questions above on allele frequencies and SV sharing. Together with the phenotypes that are available from reference 8, this would allow readers to recapitulate the heritability and association analyses. I couldn't find SNP genotypes associated with reference 8 (although I checked only briefly). These would also need to be made available to ensure that readers can reproduce the analyses presented here. If they are available somewhere, a brief mention of their location would be useful in the present paper.

Response: *All SNP, indel and structural variant calls, genotypes and copy numbers are available on figshare at: https://figshare.com/projects/fission_yeast_structural_variation/15798 Array data is available at ArrayExpress, accession number: E-MTAB-4019. All these sources of data have been described in the manuscript.*

3.5) Have you done qPCR to confirm some of the CNVs, especially those that segregate among multiple clonal populations?

Response: *No, we did not. But the expression arrays effectively confirmed the non-shared CNVs within clusters. The pairs selected are marked in Figure 2b (colored dots) and we could confirm a change in expression for all of the duplications.*

3.6) In the visual inspections for the SV calls, what types of artifacts or features did you look for? What were typical failure modes for putative SVs that you deemed incorrect? A brief description in the Methods would be useful to the community.

Response: *We utilized different visualizations from IGV and focused on the mapping reliability and overall signal for each SV type. For example for translocations, we often observed a region where the pairs of the reads mapped to multiple distinct regions/chromosomes. In this case we discarded this region. Interestingly, this was not filtered by samtools or subsequent tools since one pair was mapping uniquely. We have extended our guidelines in the method section.*

Minor comments:

3.7). Supplementary Figure S3: It would be helpful to indicate the absolute coverage of the strains as well. This would help to get a better sense of the strength of the signal. For example, a two-fold coverage difference means more with a 100X coverage baseline than a 2X baseline. If different strains had different average genome coverage, how were the relative coverages in the plots calculated? Were they anchored to the flanking sequence somehow, or are they purely "coverage strain 1 / coverage strain 2"? I'm trying to understand why some of the green strains in the figure have less coverage than the reference. The normalization scheme would probably explain this.

Response: *Before the calling and mapping we randomly subsampled reads for each strain such that we had an average theoretical coverage of 40x per sample.*

3.8) p. 14 l. 262 "Our analysis of heritability showed that SNPs are generally able to capture most of the genetic contribution of SVs" seems to contradict the result on p. 13 l.

233 that "Analysis of simulated data confirmed that the contribution of CNVs could not be explained by linkage to causal SNPs alone". Please clarify.

Response: *It is true that SNPs are generally able to capture most of the genetic contribution of SVs, but not all. However, we show via analysis of real data, and simulated data that SVs do make a contribution (averages: CNVs 7%, and rearrangements 4%). Therefore some of the heritability assigned to SNPs (when SNPs are the only variants used) – is actually due to SVs.*

3.9) p. 17 l. 311 "we found that rearrangements explained spore viability better than CNVs [...]" this implies that you tested rearrangements and CNVs directly against each other, perhaps as you did further down for SNPs and rearrangements. Please rephrase this to "while rearrangements correlated with spore viability, there was no significant correlation between CNVs and viability".

Response: *We have corrected this sentence.*

3.10) Figure 4: the legend has an incorrectly rounded p-value: SNPs | rearrangements = 0.03, whereas the figure gives $p = 0.038$, which is $p = 0.04$ after rounding. The correlation estimate is also slightly different between legend and figure.

Response: *We have corrected this, so that the figure and the text now have the same values.*

3.11) p. 19 l. 366 Instead of a "measureable" rate, do you mean "considerable" or simply "high"? All mutation rates can be measured.

Response: *We altered this sentence to: "at a rate of approximately one CNV/10 generations".*

3.12) Supplemental Figure S8: in the top left panel, in the leftmost bar, the open circle above the bar should probably be filled? If not, why is the "estimate - 1sd" higher than the estimate?

Response: *Due to our reanalysis of heritability this figure has been replaced.*

3.13) Abstract: "genomics regions" should be "genomic regions"

Response: *This has been corrected.*

Reviewer #4

This paper focuses on the effects of structural variation on phenotypic differences and reproductive isolation in *Sc. pombe*. Although the work in this manuscript is performed well, I had some significant criticisms:

4.1) Not very much was done with the ample phenotype dataset to make specific connections between genetic variants and traits.

Response: *We respectfully disagree. We estimated the heritability contribution for 227 traits with respect to SNPs, CNVs and rearrangements. We compare this to simulated data where either SNPs, CNVs or rearrangements ‘cause’ the trait (to ensure that our results were not artefacts caused by linkage). Then we conducted 227 genome-wide association studies to identify specific variants that have contributed to traits. Finally, we quantified the effects of SNPs, CNVs or rearrangements on viability of the offspring when crossing two different strains. Given the number of strains that are available and current methods of quantitative genetics, we think that these are the most appropriate analyses.*

4.2) It is known that *Sc. pombe* isolates exhibit a substantial amount of structural variation. This paper improves upon our knowledge of this details of this structural variation, but at this juncture, these details seem to represent an incremental advance.

Response: *We agree with the reviewer that we improve the knowledge of SVs compared to previous studies. Indeed genomic difference has been studied before, e.g. the work of Brown et al 2011⁹ (using pulse field gels reported 14 CNV and one inversion), Avelar et al.¹⁰ (focused on artificial introduced SVs) and Zanders et al (2014)¹¹ (focusing on rearrangements between two *Saccharomyces* strains)(we know of no others). Our data set was different, in that it contained traits, SNPs, expression data and data from crosses. See 2.1 for a list of novelties for our study. These unique features of our work provide a new and more revealing analysis than has been conducted previously.*

4.3) A large amount of work in *Sa. cerevisiae* has shown that structural variation can have important phenotypic and gene expression effects, and that some of these structural variations can be transient. I thought the attempt to determine the quantitative contribution of structural variation to phenotypic variation was of value, but the insights gained also seemed incremental.

Response: *We agree that some excellent work has been produced in budding yeast, and we have added more citations of this work. We would suggest that our analysis is distinct, and of high value, because we conducted several unique analyses that have not been published in budding yeast:*

- 1. We did a genome- and population scale discovery of rearrangement (inversion/translocation) variants, as well as CNVs.*
- 2. We showed via a quantitative analysis that SVS were transient (new analysis in response to reviewers' comments).*
- 3. We estimated the contribution of all type of variants to heritability for a large number of different types of traits.*
- 4. We conducted GWAS.*
- 5. We showed that rearrangements contribute to intrinsic reproductive isolation while controlling for the correlation of SNP-distance with SV-distance.*

See response 2.1, 4.2 and 4.7 for more details.

4.4) A number of papers in *Saccharomyces* have shown the transient nature of structural variation.

Response: *See response 2.1, 4.2 and 4.7*

4.5) Extensive work by Gianni Liti and Ed Louis on reproductive isolation in *Saccharomyces*, especially *Sa. paradoxus*, already has shown relationships between amount of structural variation and reproductive isolation. The fact that CNVs may not impact this relationship is to be expected.

Response: *Yes, and we now cite a few more of these & other yeast papers. Importantly though, we find that the structural variant genetic distance between parents is strongly*

correlated with the SNP genetic distance. So without controlling for this correlation, it is not possible to tell whether SNPs (or other small changes), and/or rearrangements cause reproductive isolation in S. pombe (we show evidence that both have an effect). With respect to CNVs, we think it valid to confirm the expected result with new data, particularly to contrast with the larger effect of CNVs on traits.

4.6) The aesthetics of the figures could be improved; e.g., Fig 3 might be better if plots with points instead of bars were used and Fig 4 might be aided by a legend panel indicating the difference between red and black points or differently sized points.

Response: *We have tried improving Fig 3, but points did not look better to our eye. The main point of part A is to show that total heritability increases when we add SVs, so we highlight that in the legend.*

4.7) It was surprising that more work from Saccharomyces was not cited. This was especially true in the section on reproductive isolation, where the work mentioned above, which arguably represents the gold standard for yeast papers on the topic, was not even recognized. Ultimately, many of the questions addressed in this paper have been extensively examined in Saccharomyces. Even though this is a different yeast genus, it is still important to cite and discuss the prior work in Saccharomyces and describe how this paper builds upon it.

Response: *Yes, there has been some excellent work produced describing structural variants in budding yeast, and we have added a few relevant citations. But, to our knowledge there is no population-wide survey of both copy number variants (large duplications/deletions, rather than small 'indels') and rearrangements (inversions/translocations), and certainly none that combined SVs with expression levels and traits. If there are any other studies that we missed, or failed to appreciate, please let us know.*

In summary, the science and writing in this paper were solid. However, this paper had insufficient novelty and awareness of historical context to warrant publication in Nature Communications.

Response:

We and seemingly the other reviewers disagree with this conclusion. As listed above (reply to 2.1 and 4.2), we highlight multiple novel findings within a single population that illustrate the varied effects of structural variants. For example, i) we conduct a new analysis in this revision that shows the transience of CNVs, and we back this up with an RNA-expression analysis which shows clear effects on expression ii) different types of SVs impact reproductive isolation differently (the lack of effects of CNVs has been assumed, but not demonstrated before to our knowledge) iii) CNVs have more impact on traits than rearrangements.

*The *S. pombe* community will profit from this analysis by having a highly accurate genome and population wide catalog of SVs, where the sensitivity is comparable to the 1000 genome project paper (~76% verified by PCR). Particularly with this species being used for more quantitative genetic studies that require crosses (we are aware of this because we have distributed the strains to many laboratories after our previous Nature Genetics paper).*

Moreover, for the field of computational biology our SURVIVOR tool provides useful novel methods to: i) compare structural variation calls (the tool is currently applied in multiple projects e.g. Genome in the Bottle that aims to establish gold standard set of SV using multiple technologies) ii) to simulate SVs to aid in evaluation of SV-callers, not only on simple events but also for complex events that observed in human cancer (chromoplexy and chromothripsis). This matters because the optimal caller for their particular data set is far from clear at the moment.

References

1. Bergström, A. *et al.* A high-definition view of functional genetic variation from natural yeast genomes. *Mol Biol Evol* **31**, 872–888 (2014).
2. Jeffares, D. C. *et al.* The genomic and phenotypic diversity of *Schizosaccharomyces pombe*. *Nature Genetics* **47**, 235–241 (2015).
3. Speed, D., Hemani, G., Johnson, M. R. & Balding, D. J. Improved Heritability Estimation from Genome-wide SNPs. *Am. J. Hum. Genet.* **91**, 1011–1021 (2012).
4. Carr, A. M., MacNeill, S. A., Hayles, J. & Nurse, P. Molecular cloning and

sequence analysis of mutant alleles of the fission yeast *cdc2* protein kinase gene: implications for *cdc2+* protein structure and function. *Mol Gen Genet* **218**, 41–49 (1989).

5. Chan, J. E. & Kolodner, R. D. A genetic and structural study of genome rearrangements mediated by high copy repeat Ty1 elements. *PLoS Genet* **7**, e1002089 (2011).
6. Vinces, M. D., Legendre, M., Caldara, M., Hagihara, M. & Verstrepen, K. J. Unstable tandem repeats in promoters confer transcriptional evolvability. *Science (New York, NY)* **324**, 1213–1216 (2009).
7. Gadaleta, M. C. *et al.* Swi1 Timeless Prevents Repeat Instability at Fission Yeast Telomeres. *PLoS Genet* **12**, e1005943 (2016).
8. Coulon, S. *et al.* Slx1-Slx4 are subunits of a structure-specific endonuclease that maintains ribosomal DNA in fission yeast. *Mol. Biol. Cell* **15**, 71–80 (2004).
9. Brown, W. R. A. *et al.* A geographically diverse collection of *Schizosaccharomyces pombe* isolates shows limited phenotypic variation but extensive karyotypic diversity. *G3* **1**, 615–626 (2011).
10. Teresa Avelar, A., Perfeito, L., Gordo, I. & Godinho Ferreira, M. Genome architecture is a selectable trait that can be maintained by antagonistic pleiotropy. *Nat Commun* **4**, 2235 (2013).
11. Zanders, S. E. *et al.* Genome rearrangements and pervasive meiotic drive cause hybrid infertility in fission yeast. *eLife* **2014**, e02630 (2014).

REVIEWERS' COMMENTS:

Reviewer #1 (Remarks to the Author):

The Authors have thoroughly addressed my previous comments, and I have no further ones to make. I defer to other Reviewers in regard to novelty of the findings in *S.pombe*, as this is not my area of expertise.

Reviewer #2 (Remarks to the Author):

I am satisfied with the response to my comments - the text changes now better distinguish this study from previous work in the other yeast + human populations.

Reviewer #3 (Remarks to the Author):

Thanks to the authors for carefully addressing all my previous comments. I have just a few minor remaining comments:

1. p. 9 l. 160 refers to Figure 3c, which does not seem right. Should this be Figure 2c or some other Figure?

Response: *Yes, there was an error here. Text changed to:* Notably, this collection showed four non-overlapping segregating duplications (**Fig. 2c**, yellow highlight). This striking finding suggests that CNVs can arise or disappear frequently during evolution.

2. p. 9 l. 165: “strong correlation between the total mutation in these regions and the total variation in copy number of the CNV” is awkwardly phrased. “total mutation” sounds like it includes the CNV, which seems wrong. Please reword.

Response: *Text changed to:* If a CNV was subject only to the same processes as these adjacent regions, we would expect a strong correlation between the rate of **point mutation (SNPs)** in these regions and the total variation in copy number of the CNV.

3. Supplementary Figure 4 a & b share one axis, but in a) it is the x-axis while in b) it is on the y axis. Please make this consistent.

Response: *This has been changed so they two plots have a consistent axis.*

4. There are two each of Supplementary Figures 6 and 7. Please fix.

Response: *This has been corrected.*

5. In the second Supplementary Figure 7, the top middle and top right panels have the same axis labels, but show different data. Please clarify.

Response: *This has been corrected, the first plot was all SV differences, while the second only counted rearrangement differences.*

Reviewer #4 (Remarks to the Author):

The revised version of this manuscript represents a significant improvement over the initial submission. The authors do a much better job now of connecting their work to previous papers from other groups, including labs that work on *Saccharomyces cerevisiae*.

It is clearer how the components of the paper collectively build into a manuscript that could be of value to a number of different groups of researchers (e.g., people working on *S. pombe*, quantitative genetics, folks interested in structural variation).

Response: *We appreciate these positive comments, and we agree.*

Aside from one comment, I am satisfied with how the authors addressed my remarks and handled input from the other reviewers. However, there was a misinterpretation of my first point, perhaps because I could have been clearer: ‘Not very much was done with the ample phenotype dataset to make specific connections between genetic variants and traits’. What was meant by this point is that the authors do not discuss how any specific variants influence any specific traits? In other words, no discussion of the molecular and systems mechanisms contributing to heritable phenotypic variation in this organism is provided. For example, on p15, the authors write: ‘Thus, some groups of traits have consistently larger contributions from SVs than from SNPs alone. These traits include intracellular amino acid concentrations...’ Can you make any connection to the mechanisms based on SNVs? This seems especially feasible for CNVs, which are often resolved to individual genes. There are other similar opportunities in the paper. I don’t think these modifications are absolutely necessary, but they would certainly help make this paper more accessible to researchers who are not statistical geneticists.

Response: *We agree that the work was to some extent best suited to statistical geneticists, as is the data. We did attempt to specify a vignette in the previous version, but on closer analysis this was shown to be an artifact. Therefore we are wary of making claims that are difficult to substantiate without laboratory verification. We ourselves prefer the perspective gained from the analysis of the effects of all structural variants (or separated into CNVs and rearrangements) because we find this overview more revealing. We hope that readers will agree.*

More minor comments:

Yeast species: Often in a context where both *Saccharomyces cerevisiae* (or related species) and *Schizosaccharomyces pombe* are being discussed, the former and latter will be referred to as *Sa. cerevisiae* and *Sc. pombe*, respectively, to prevent confusion.

Response: *We altered the text to use *Sa. Paradoxus* in the one instance we mentioned budding yeasts, but since most of the paper is about *S. pombe*, we kept this as it was.*

P3, 168: The word ‘progress’ read weird to me. Maybe ‘advance’?

Response: *We agree, changed to advance.*

P4, 171: ‘Various aspects of biology’ is a vague phrase.

Response: *Changed to “a variety of quantitative traits and intrinsic reproductive isolation.”*

P21, 1398-402: In these sentences the authors mention experimental studies in budding yeast, but then an example from *S. pombe* is provided.

Response: *Changed to “Our analysis is consistent with experimental studies with fission yeast”.*